**Subject Category:**
Biology (whole organism)

neuroscience/evolution

sea snake, scale organs, cutaneous, mechanoreceptor, skin, sensilla

**Author for correspondence:**
Jenna M. Crowe-Riddell
e-mail: jmcroweriddell@gmail.com

# Ultrastructural evidence of a mechanosensory function of scale organs (sensilla) in sea snakes (Hydrophiinae)

Jenna M. Crowe-Riddell[1], Ruth Williams[2],
Lucille Chapuis[3] and Kate L. Sanders[1]

[1] School of Biological Sciences, The University of Adelaide, Adelaide, South Australia 5005, Australia
[2] Adelaide Microscopy, the Centre for Advanced Microscopy and Microanalysis, Adelaide, South Australia 5005, Australia
[3] College of Life and Environmental Science, University of Exeter, Exeter EX4 4QD, UK

JMC-R, 0000-0003-2794-2914; LC, 0000-0003-3001-983X;
KLS, 0000-0002-9581-268X

The evolution of epidermal scales was a major innovation in lepidosaurs, providing a barrier to dehydration and physical stress, while functioning as a sensitive interface for detecting mechanical stimuli in the environment. In snakes, mechanoreception involves tiny scale organs (sensilla) that are concentrated on the surface of the head. The fully marine sea snakes (Hydrophiinae) are closely related to terrestrial hydrophiine snakes but have substantially more protruding (dome-shaped) scale organs that often cover a larger portion of the scale surface. Various divergent selection pressures in the marine environment could account for this morphological variation relating to detection of mechanical stimuli from direct contact with stimuli and/or indirect contact via water motion (i.e. 'hydrodynamic reception'), or co-option for alternate sensory or non-sensory functions. We addressed these hypotheses using immunohistochemistry, and light and electron microscopy, to describe the cells and nerve connections underlying scale organs in two sea snakes, *Aipysurus laevis* and *Hydrophis stokesii*. Our results show ultrastructural features in the cephalic scale organs of both marine species that closely resemble the mechanosensitive Meissner-like corpuscles that underlie terrestrial snake scale organs. We conclude that the scale organs of marine hydrophiines have retained a mechanosensory function, but future studies are needed to examine whether they are sensitive to hydrodynamic stimuli.

# 1. Introduction

Hardened epidermal scales are a characteristic trait of snakes (and other lepidosaurs: lizards and tuatara) that facilitate defensive signalling, camouflage, water retention and locomotion [1–3]. The epidermal scales also provide the primary surface for mechanoreception, which is the ability to sense mechanical stimuli that result from physical displacement (vibration) [4]. Scale organs known as 'sensilla' or 'tubercles' *sensu* [5–7] are small mechanoreceptors that protrude from the surface of epidermal scales of the head and body of snakes. Snakes are likely to use these mechanosensory organs to explore and navigate substrate [8,9], and during courtship [10] and feeding [11,12] behaviours. However, the anatomy and neurophysiology of scale organs are conspicuously understudied in comparison to other sensory organs, for example, eyes [13], auditory structures [14], vomeronasal organ [15] and heat-sensing pits [16].

In terrestrial snakes, scale organs are concentrated on the head and are highly sensitive to mechanical stimulation, particularly moving stimuli [17–20]. The underlying ultrastructure of cephalic scale organs consists of an innervated cluster of dermal cells ('dermal papilla') that displaces the surrounding epidermis to create round skin elevations [21–23]. These underlying features of scale organs have been likened to 'Meissner corpuscles', which are low-threshold mechanoreceptors (LTMRs) sensitive to innocuous 'light touch' stimuli in the glabrous (hairless) skin of mammals [24,25]. Scale organs on the head are more specialized in their underlying ultrastructure than on the body, which lack dermal papillae and have outer skin elevations that are instead caused by a superficial thickening of the epidermis [10,26].

Snakes exhibit substantial variation in the size, shape, density and distributions of their scale organs. Enlarged and/or high densities of scale organs have been reported in fossorial snakes (e.g. Leptotyphopidae) and some sea snakes (Hydrophiinae), whereas in other colubroid snakes scale organs are small and/or sparse (e.g. Dipsadinae) or even absent in some species (e.g. Viperidae) [22,27–29]. Interspecific differences in the traits of scale organs probably relate to various aspects of species' environment, ecology and phylogeny. However, our understanding of the adaptive diversity of snake scale organs is hindered by a lack of comparative data describing differences in the external traits of scale organs and their underlying ultrastructure.

Hydrophiine snakes (Elapidae) provide a useful comparative framework to investigate the evolution of squamate scale organs in response to major ecological transitions [7]. The viviparous sea snakes comprise a clade of more than 60 species that evolved within the terrestrial Australian hydrophiine radiation (tiger snakes, death adders, taipans) approximately 9–18 Ma [30]. Previous work has found that the cephalic scale organs of sea snakes are substantially more protruding (dome-shaped) compared to their terrestrial counterparts, and in some lineages cover a much larger portion of the scale surface (greater than 6% versus less than 2.5% in sampled taxa) [7]. This divergence in external morphology might reflect divergent selection pressures in the marine environment. However, the hitherto lack of data on the ultrastructure of scale organs in sea snakes precludes meaningful comparisons with terrestrial snakes.

In their external appearance, the dome-shaped scale organs of sea snakes closely resemble the integumentary scale organs (ISOs) of crocodilians, which are cephalic mechanoreceptors with elaborate Merkel cell-neurite complexes and sensitivity to water motion (i.e. hydrodynamic reception) [31–33]. A dome-shaped scale organ provides increased surface area for stimuli to be received from multiple directions, possibly enhancing mechanoreception in an aquatic habitat whereby water motion can be detected from both biotic sources, e.g. conspecifics, prey and predators, and abiotic sources, e.g. turbulence caused by water currents deflected past objects [4]. Indeed, two independently aquatic snakes, *Erpeton* and *Acrochordus*, are distantly related to hydrophiine sea snakes but have protruding organs that are likely to be sensitive to water motion generated by the movement of fish prey [5,34]. It is also plausible that scale organs have been co-opted in sea snakes for a different sensory modality, such as dermal phototaxis found in *Aipysurus* sea snakes [35,36], or electromagnetic sensing for navigation [15]. Alternatively, scale organs may have been co-opted for a non-sensory function such as enhanced friction for gripping during mating, or disruption of the skin boundary layer to increase swimming performance (analogous to the denticles of shark skin or tubercles on the fins of whales [37–39]).

We aimed to better understand the evolution of scale organs in sea snakes by describing their ultrastructure in two fully aquatic species, *Aipysurus laevis* and *Hydrophis stokesii*, using immunohistochemistry, and light and electron microscopy. If sea snake scale organs have a mechanosensory role, either close-contact touch or detection of water motion (deflected off objects or

prey/predators), we would expect them to have retained the ultrastructure described in terrestrial snakes, and possibly contain other sensory cells such as the Merkel cell-neurite complexes of crocodilian ISOs. Co-option for alternative sensory roles would be implicated if different cell types are present. Finally, if dome-shaped scale organs provide a non-sensory (e.g. structural) function, we would expect their elevation from the skin surface to be created by superficial thickening of the epidermis with no associated neuronal or receptive cells.

# 2. Material and methods

## 2.1. Specimens and tissue sampling

Two museum specimens of the sea snake species *Aipysurus laevis* (one individual) and *Hydrophis stokesii* (one individual) were used for gross morphological observations. Fresh specimens of these species (two individuals of *A. laevis*; one individual of *H. stokesii*) were collected 1–10 km offshore from the coast of Broome, Western Australia, in June 2015 and September 2016.

Immediately after euthanasia, cephalic scales were sampled from all three sea snakes, and tail scales were sampled from the posterior dorsal surface and ventral tip of the tail in a single *A. laevis* because this species exhibits tail phototaxis linked to dermal photoreception [36]. Entire scales were dissected to sample the whole skin from epidermis to subcutaneous tissue. The specimen details and locations of sampled scales are shown in table 1 and figure 1. A single specimen of *Oxyuranus scutellatus* (the Australian taipan) was sourced from a captive breeding population (Venom Supplies Pty Ltd, South Australia) to sample brain tissue for antibody controls (see below), because this species is closely related to viviparous sea snakes [30]. All samples were fixed by immersion in either 4% paraformaldehyde for immunohistochemistry, or 1.5% glutaraldehyde and 4% paraformaldehyde for electron microscopy. After immersion in fixative for 24 h, samples were washed and stored in phosphate buffered saline (PBS; pH 7.4) with sucrose, before being transferred into phosphate buffer with 0.05% sodium azide.

## 2.2. Stereo and light microscopy

The outer skin morphology of museum specimens was examined using a stereomicroscope with a mounted camera (SMZ25, Nikon Inc., Japan). Specimens were submerged in water and illuminated by a ring of light-emitting diodes (P2-FIRL LED Ring Illumination Unit, Nikon Inc., Japan) to reduce specular reflections from the scales. A high-depth-of-field photographic image was composed using imaging software (NIS-Elements Advanced Research v. 5.10, Nikon Inc., Japan).

The general cellular morphology of the skin samples was examined using light microscopy. Samples were dehydrated by successive immersion in alcohol, then paraffin-embedded for serial sectioning (10 μm). Slides were stained with hemotoxylin-eosin or Gomori's one-step trichrome [41], scanned using a digital slide scanner (Nanozoomer, Hamamatsu Photonics, Japan) and measurements taken using imaging software (Nanozoomer Digital Pathology v. 2.6, Hamamatsu Photonics, Japan). We measured the thickness of the epidermis located above scale organs, and at adjacent areas of skin that did not contain organs. Because the outer layer of hardened skin (beta layer) sometimes became artificially separated from surrounding layers during tissue processing, we measured only the living (nucleated) epidermal layer (*stratum germinativum*). The diameter and height of dermal papillae and other dermal structures were measured and the ratio of diameter : height calculated.

## 2.3. Statistics

We used the two-sample *t*-test (unpaired) to examine differences in epidermal thickness between scale organs and adjacent skin that did not contain scale organs. Before statistical analyses, we checked that data were normally distributed using Bartlett's test. Statistical analyses were performed using base packages in R v. 3.5.1 [42].

## 2.4. Immunohistochemistry

Immunohistochemistry was performed on paraffin-embedded serial sections (10 μm) for a neuronal marker, protein gene product 9.5 (PgP9.5). Briefly, slides were blocked for endogenous peroxidase

**Table 1.** Taxonomy, life stage, museum accession or field numbers and sample size of two species of sea snakes (Hydrophiinae) used in this study. Tissue samples were collected for various microscopy analyses: stereomicroscopy (SM), light microscopy (LM), transmission electron microscopy (TEM) and immunohistochemistry (IHC). Museum specimens were sourced from the Western Australian Museum (WAM) and the Field Museum of National History, Chicago (FMNH).

| taxonomy | | specimen information | | | | tissue samples | microscopy analyses |
|---|---|---|---|---|---|---|---|
| genus | species | museum or field numbers | sex | life stage | time since last shed skin (days) | scale type and location | |
| Aipysurus | laevis | KLS0690 | M | adult | 18 | sixth supralabial (right side) | LM |
| | | | | | | posterior tip of tail (right side) | LM |
| | laevis | #AL270916 | M | adult | >128 | nasal scale (right side) | TEM |
| | laevis | WAMR174260 | M | subadult | unknown | gross morphology of skin | SM |
| Hydrophis | stokesii | #HS270916A | M | adult | 107 | nasal scale (right side) | LM & IHC |
| | | | | | | nasal scale (left side) | TEM |
| | stokesii | FMNH202826 | unknown | juvenile | unknown | gross morphology of skin | SM |

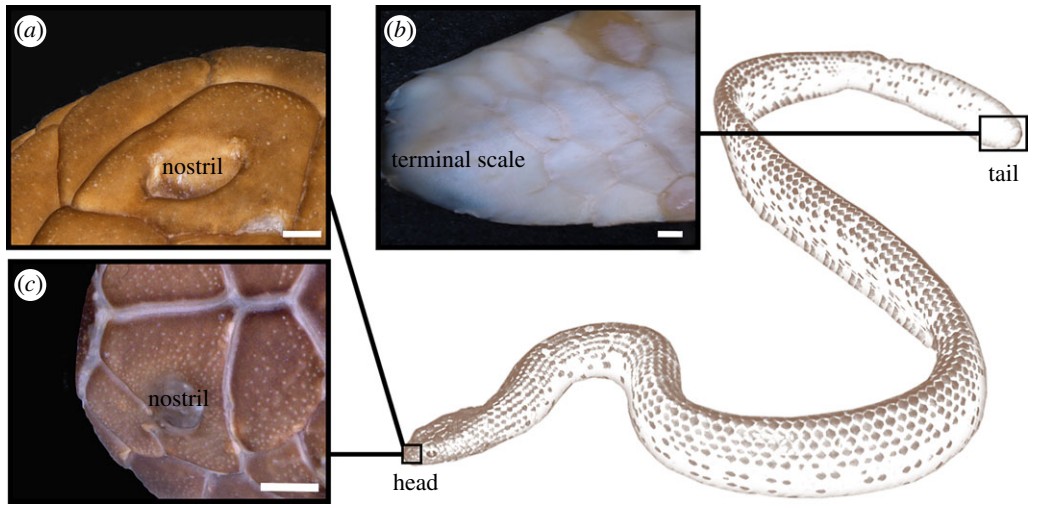

**Figure 1.** Gross morphology of the skin of sea snakes illustrating small, unpigmented scale organs (sensilla). Line drawing of sea snake indicates regions of skin sampled for this study: nasal scales from the head of *Aipysurus laevis* and *Hydrophis stokesii*, and supralabial scales from the head and caudal scales from the tail in *A. laevis* only. (*a*) Gross morphology of scale organs on the nasal scale of *A. laevis*. (*b*) Gross morphology of the caudal scales of *A. laevis* illustrating sparse scale organs. (*c*) Gross morphology of scale organs on the nasal scale of *H. stokesii*. Stereomicroscope images were taken from museum specimens: (*a,b*) WAMR174260 and (*c*) FMNH202826. Scale bars represent 1 mm. Line drawing based on the image of *A. laevis* from [40] and modified with permission.

with 0.5% hydrogen peroxide in methanol at room temperature for 30 min. Slides were rinsed in PBS and processed in 10 mM sodium citrate (pH 6.0) for heat-induced epitope retrieval. Slides were washed twice in PBS, before blocking in 3% normal horse serum (NHS) in PBS for 30 min. Sections were incubated with mouse monoclonal anti-PgP9.5 antibody (dilution 1 : 2000 with 3% NHS) at room temperature overnight. Sections were then washed twice in PBS and incubated with a peroxidase-conjugated secondary antibody (IgG anti-mouse, 1 : 500 diluted in PBS with 3% NHS) for 30 min, then incubated with streptavin peroxide (dilution 1 : 1000 with 3% NHS) for 1 h. Binding sites were revealed using a red chromogen (NovaRed Peroxidase Substrate Kit, Vector, USA) according to the manufacturer's instructions and incubated for 2–3 min. Slides were washed in distilled water for 5 min before counterstaining in Harris haematoxylin for 30–60 s and allowed to air dry. A primary antibody control was performed using the above protocol on snake (taipan) brain tissue; a secondary antibody control was performed using the above protocol, with the primary antibody incubation step omitted, on snake brain and cephalic skin tissue. Slides were imaged using an optical microscope (BX51, Olympus, Australia) and the saturation and hue of images was adjusted using imaging software (Adobe Photoshop v. 2017.1.1, Adobe Systems Inc., USA). Unfortunately, due to preservation issues, we were unable to perform immunohistochemistry on these cephalic skin sections in *A. laevis*.

## 2.5. Electron microscopy

To view ultrastructure, skin samples were prepared for electron microscopy. Samples were post-fixed in 2% osmium tetroxide solution, then dehydrated in ascending series of ethanol and infiltrated in epoxy resin. Resin blocks were then polymerized overnight at 70°C. Semi-thin (1 μm) sections were cut and stained with toluidine blue to locate an individual scale organ under light microscopy. Ultra-thin (70 nm) sections were cut and stained with uranyl acetate and lead citrate. Sections were placed on nickel-coated mesh grids and viewed at 100 kV under a transmission electron microscope (Tecnai G$^2$ Spirit TEM, FEI Company, USA).

## 3. Results

Several epidermal layers were identified using light microscopy: the nucleated layer (*stratum germinativum*) was non-corneous and consisted of a basal layer of elongate or columnar cells and one to three layers of round, loosely arrange keratinocytes; the non-nucleated layer (*stratum corneum*) consisted of corneous α and β cells. According to definitions from [1,3], skin samples that were

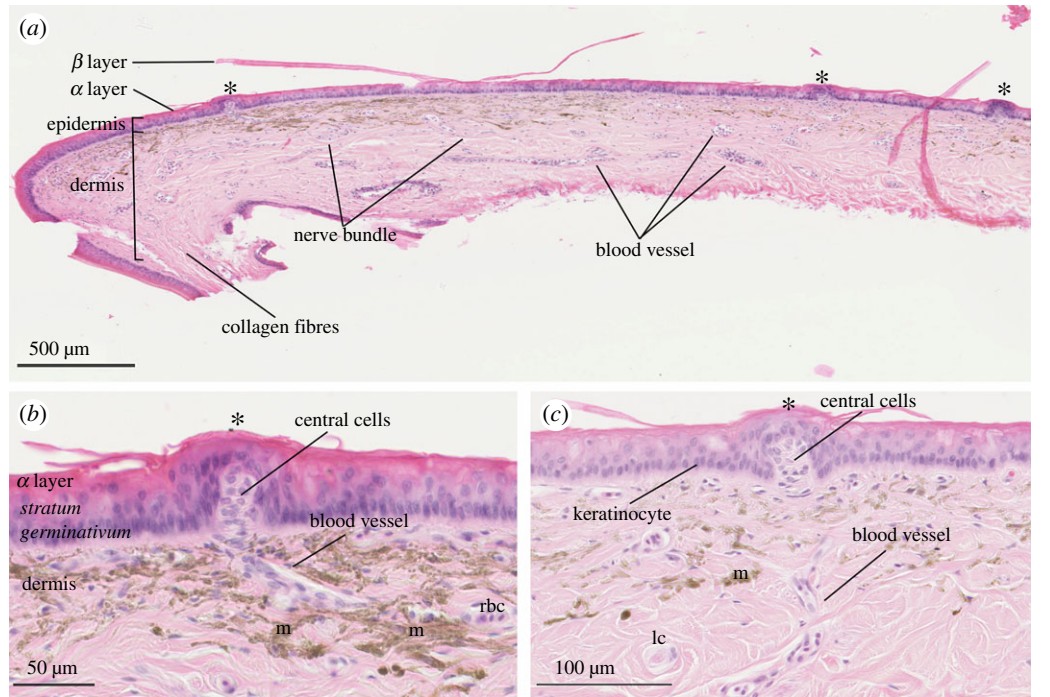

**Figure 2.** Light micrographs of a transverse section of cephalic skin (supralabial scale) from *Aipysurus laevis*. (*a*) Transverse section shows that scale organs (\*) are skin elevations (bumps) created by dermal papillae; other features of the dermis are clearly visible including nerve bundles, blood vessels and collagen; note that the beta layer has artificially separated from the alpha layer. (*b,c*) Higher magnification of transverse section of scale organs (\*) that show central cells within the dermal papilla, which displace the *stratum germinativum* of the epidermis; dermal papillae are vascularized by blood vessels; note the red blood cells (rbc), lamellar corpuscles (lc) and melanophores (m) within the dermis. Slides were stained with haematoxylin-eosin and magnified at (*a*) ×5.5, (*b*) ×20 and (*c*) ×30.

viewed under light microscopy (table 1) were in the resting phase of the epidermal shedding cycle; skin samples viewed under the electron microscope (table 1) appeared to be in pre-renewal phase.

## 3.1. Cephalic scale organs

Observed under a stereomicroscope, the cephalic scale organs appeared as unpigmented external elevations (bumps) of outer skin (figure 1). Observed under light microscopy, the cephalic scale organs of *A. laevis* (figure 2) and *H. stokesii* (figure 3) shared a similar structure that consisted of a cluster of 9–11 cells (central cells), originating in the dermis and evaginating the epidermis to create a dermal papilla. The ratio of length to diameter of the dermal papilla was approximately 1 : 1 for both *A. laevis* and *H. stokesii* (electronic supplementary material, table S1). The dermal papilla was occasionally tapered at its basal end in *H. stokesii* (figure 3*a*), but this is likely to be an artefact of the tissue sectioning. In some dermal papillae, we were able to identify a blood vessel leading to (and thus presumably vascularizing) the central cells (figure 2*b*). In *H. stokesii*, the Gomori's one-step trichrome stain revealed collagen fibres interspersed between central cells and often separated the dermal papilla from keratinocytes within the epidermis (figure 3*c*). In both species, the dermal papilla displaced surrounding epidermal layers so that the columnar cells of the *stratum germinativum* were positioned above the dermal papilla, causing the bumps of the outer skin surface (figures 2 and 3). In *A. laevis*, the epidermis above the dermal papilla was approximately 50% thinner than the epidermis of the surrounding regions of skin that did not contain organs (17 μm; $t = -11.16$, 110 d.f., $p < 0.001$) and in *H. stokesii*, it was approximately 15% thinner than the adjacent flat epidermis (28 μm; $t = -2.19$, 67 d.f., $p = 0.03$).

There was a second type of dermal papilla on the cephalic scales in *H. stokesii* that contained approximately 10 central cells and displaced the surrounding epidermis but, in contrast to the cephalic scale organs, did not result in a distinctive bump in the outer skin surface (figure 4). These smaller scale organs were more variable in shape compared to typical organs (ratio length : diameter 1.7; electronic supplementary material, table S1) and often located at the base of depressions on the

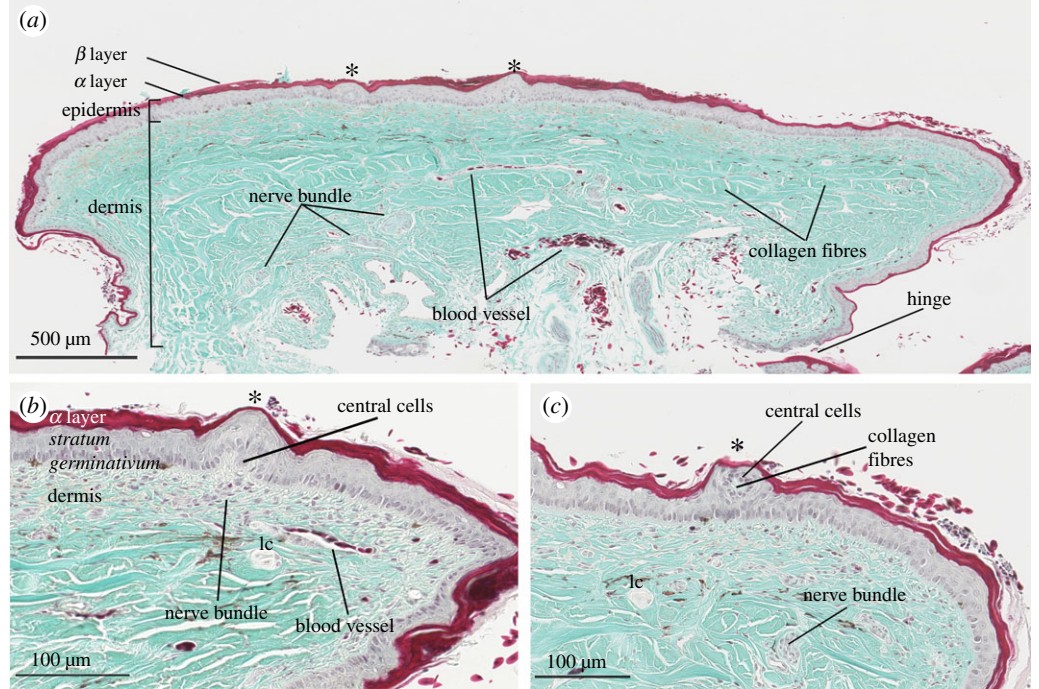

**Figure 3.** Light micrographs of a transverse section of cephalic skin (nasal scale) from *Hydrophis stokesii*. (*a*) Transverse section shows that scale organs (*) are skin elevations (bumps) created by dermal papillae; other features of the dermis are clearly visible including nerve bundles, blood vessels and collagen fibres, and hinge region of the scale. (*b*) Higher magnification of transverse section of scale organ (*); the central cells within the dermal papilla displace the *stratum germinativum* of the epidermis. (*c*) Transverse section of edge of scale organ shows a small bundle of collagen fibres surrounded by central cells. Note the lamellar corpuscles (lc) within the dermis. Slides were stained with Gomori's one-step trichrome and magnified at (*a*) ×6.2, (*b*) ×22.8 and (*c*) ×23.

outer surface of the scale (figure 4). The epidermis above the dermal papilla was 25% thinner than adjacent flat epidermis (25 µm, $t = -2.76$, 26 d.f., $p = 0.01$; approximately same height as the epidermis above papilla of other cephalic organs, $t = 0.85$, 12 d.f., $p = 0.41$).

The cephalic dermis and epidermis of *H. stokesii* were immunoreactive for PGP9.5 (figure 5). Specificity of immunoreactions was confirmed by antibody controls (electronic supplementary material, figure S1) and by the localized staining of nerve bundles that had previously been identified under light microscopy (figures 2*a* and 3*a*). Dermal axons travelled to the scale organs (figure 5*c*), then meandered through the central dermal papilla before innervating the outer epidermis and presumably terminate as distinct discoid endings in the alpha layer (figure 5*a,b*). These discoid endings were primarily located above the dermal papilla, but were also present in flat epidermis that did not contain organs (figure 6*a*). Unfortunately, the second type of dermal papillae in *H. stokesii* (described above; figure 4) was not present in the sections stained for immunohistochemistry.

Immunoreactions were also localized to ovoid structures within the cephalic dermis of *H. stokesii* (figure 6). These structures corresponded to lamellar cells that were ovoid in shape and resembled small Pacinian-like corpuscles (mean length $29 \pm 15$ µm and mean diameter of $22 \pm 12$ µm; electronic supplementary material, table S1). The location of these 'lamellar corpuscles' in *H. stokesii* ranged from 61 to 124 µm (mean 93 µm) depth from the basal layer of the epidermis. Lamellar corpuscles were also identified in the cephalic dermis of *A. laevis* that were a similar ovoid shape (mean length $37 \pm 26$ µm and mean diameter $25 \pm 5$ µm; figure 2*c*) to those found on the cephalic dermis of *H. stokesii*. The location of the lamellar corpuscles in *A. laevis* ranged from 53 to 168 µm (mean 118 µm; electronic supplementary material, table S1) depth from the basal layer of the epidermis. Although the lamellar corpuscles were dispersed throughout the dermis (*stratum laxum*), they were often subjacent to scale organs (figures 2*c* and 3*b,c*). Unfortunately, due to preservation issues, we were unable to perform immunohistochemistry on these cephalic skin sections in *A. laevis*.

The dermal papilla of a scale organ was observed in *A. laevis* using electron microscopy (figure 7). High magnification images showed a cluster of central cells within the dermal papilla (figure 7*b*). These central cells were distinguished from surrounding keratinocytes by their round shape and lack

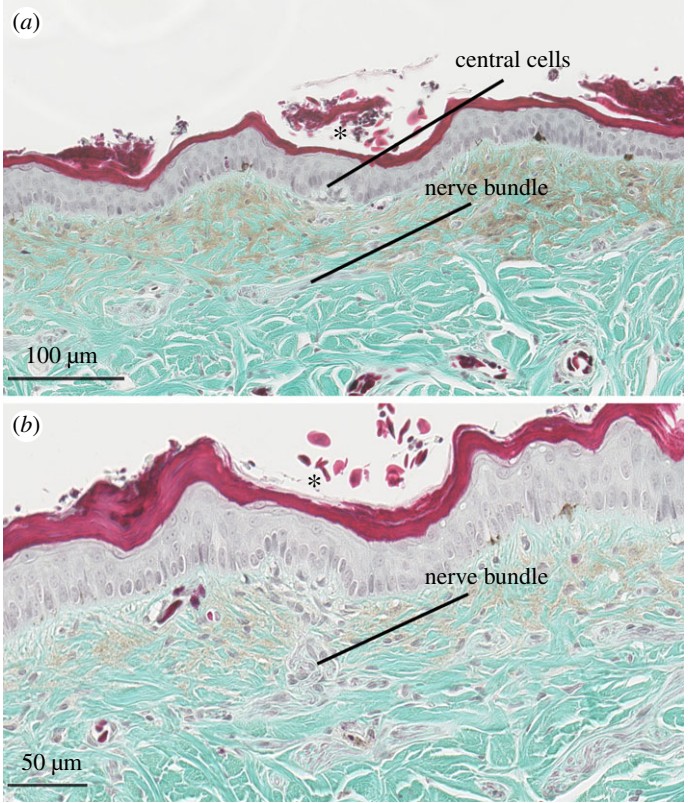

**Figure 4.** Light micrographs of a transverse section of cephalic skin (nasal scale) from *Hydrophis stokesii* showing that dermal papillae are not associated with external skin elevations (bumps). (*a*,*b*) Central cells of a dermal papillae (*) displace surrounding *stratum germinativum* of the epidermis, but do not result in skin elevations. The nerve bundle is closely associated with base of the dermal papilla. Slide was stained with Gomori's one-step and magnified at (*a*) ×20 and (*b*) ×40.7.

of tonofilaments (figure 7*b*, inset two). Tonofilaments were present in the intracellular space of keratinocytes throughout the epidermis (figure 7*b*). Tight junctions (desmosomes) associated with tonofibrils can be seen between central cells and keratinocytes (figure 7*b*, inset two). In the intercellular domain, small bundles of transverse collagen fibres and a single, small putative nerve axon were present at base of the dermal papilla (closer to dermis; figure 7*b*, inset one). Small phospholipid inclusions were also present (figure 7*b*, inset one). Unfortunately, we were unable to image the putative axon at higher magnification so could not confirm the presence of neuronal elements (e.g. lamellar arrangement of Schwann cells, neurofilaments).

## 3.2. Scale organs on the tail of *Aipysurus laevis*

Two scale structures were identified in the tail skin of *A. laevis*. We were unable to discern bumps in the outer tail skin surface using a stereomicroscope (figure 1*b*), but several skin elevations were identified in cross-sections of the skin under light microscopy (figure 8). The epidermal elevations of the tail (tail scale organs) lacked the dermal papillae associated with the cephalic scale organs. The outer bumps were instead created by thickening of the epidermis (figure 8*a*), which was 57% thicker than adjacent flat epidermis (47 μm, $t = 14.18$, 86 d.f., $p < 0.001$) and 17 μm (65%) thicker than the epidermis above cephalic scale organs ($t = -14.26$, 18 d.f., $p < 0.001$). Tail scale organs also lacked the collagen fibres and blood vessels that were associated with cephalic scale organs. A second scale structure identified in the tail skin of *A. laevis* consisted of a small dermal papilla of approximately 10 central cells with a ratio of length and diameter of 1 : 1 (figure 8*b*; electronic supplementary material, table S1). Although the dermal papilla displaced the surrounding epidermal layer (including the columnar cells of the *stratum germinativum*), this did not result in elevations of the outer epidermis (figure 6*b*). The epidermis above the dermal papilla was 42% thinner than adjacent epidermis that did not contain dermal papillae (11 μm; $t = -4.65$, 86 d.f., $p < 0.001$) and slightly thinner (5.5 μm) than the epidermis above cephalic scale organs ($t = 2.59$, 18 d.f., $p = 0.02$). Subjacent to these tail dermal papillae, collagen

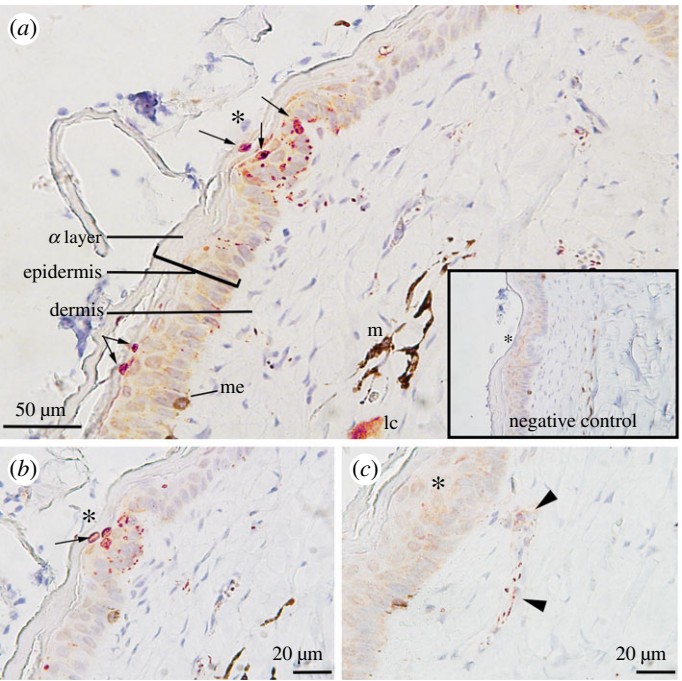

**Figure 5.** Immunoreactivity of a neuron-specific protein (PGP9.5) on cephalic skin (nasal scale) of *Hydrophis stokesii*; reactive protein appears dark pink. (*a*) Transverse section of a scale organ (*) with neuronal-positive stain within the dermal papillae, as well as within the epidermis and alpha layer above the dermal papillae. Several neuronal-positive, discoid endings (arrows) are present within the *stratum germinativum* and alpha layers of the epidermis. Lamellar corpuscles (lc) within the dermis are also immuno-positive and can be distinguished from melanocytes (me) and dispersed melanophores (m), which have a dark brown coloration. (*b*) Transverse section of a scale organ showing neuronal-positive discoid endings (arrow). (*c*) A trail of neuronal-positive stain (arrow heads) leading to a forming scale organ (*). Negative control was conducted by omitting primary antibody. Slides were counterstained with Harris haematoxylin and magnified at (*a*) ×30, (*b*) ×50 and (*c*) ×50.

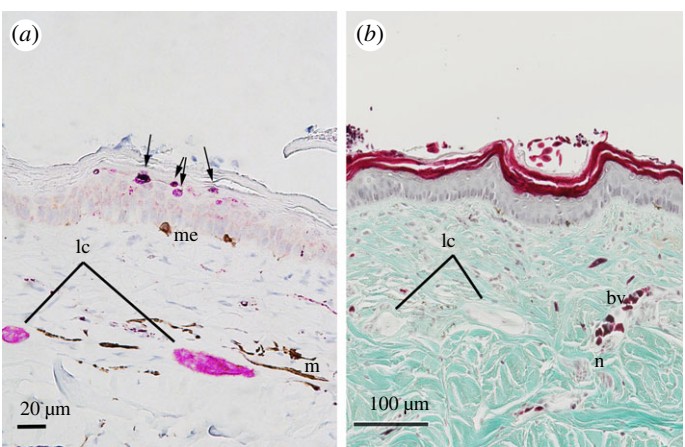

**Figure 6.** Immunoreactivity of a neuron-specific protein (PGP9.5) of lamellar corpuscles (lc) in the cephalic dermis (nasal scale) of *Hydrophis stokesii*. The location within the dermis and co-localization of immuno-staining with lamellar structures suggests that they are Pacinian-like corpuscles. (*a*) Immunoreactivity of PGP9.5; reactive protein appears dark pink, showing immuno-positive stain localized to lamellar corpuscles (lc) in the dermis and discoid endings (arrows) in the epidermis. These structures can be distinguished from melanocytes (me) and dispersed melanophores (m), which have a dark brown coloration. (*b*) Transverse section of the skin showing structure of lamellar corpuscles and an associated blood vessel (bv) and nerve bundle (n). Slides were stained and magnified: (*a*) Harris haematoxylin, ×30, and (*b*) Gomori's one-step trichrome, ×50.

fibres in the dermis (*stratum laxum*) were dispersed and melanosomes could not be seen (figure 8*b*). Unfortunately, due to preservation issues, we were unable to perform immunohistochemistry on these tail sections.

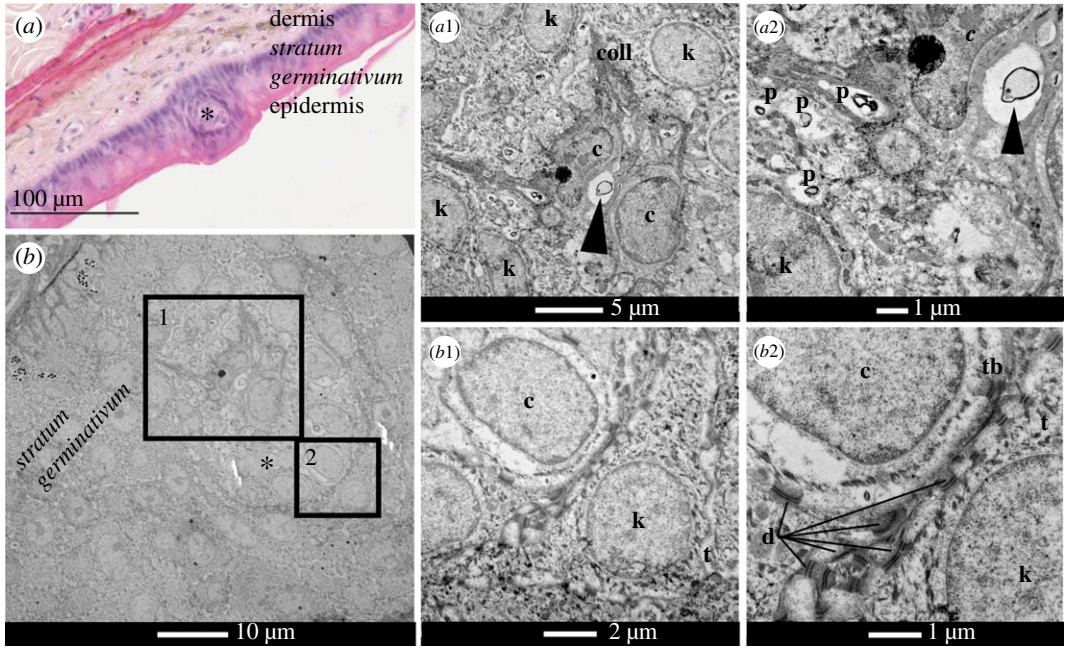

**Figure 7.** Light micrograph and transmission electron micrographs (TEM) of sections of cephalic scale organs in sea snakes. (a) Transverse section of scale organs (*) in *Aipysurus laevis* showing the dermal papilla within the epidermis. (b) Higher magnification of dermal papilla (*) in *A. laevis*. First inset shows (a1) nuclei of central cells (c) and epidermal cells (keratinocytes; k), and collagen fibres (coll), a structure typically found within the dermis, in the intercellular domain of the dermal papilla. (a2) a putative myelinated axon (arrow heads) is present in the intercellular domain of the central cells; small phospholipid (p) inclusions are also present. Inset two (b1 and b2) shows intercellular junctions (desmosomes; d) at the membrane of central cells (c) and the keratinocytes (k). Note the fine keratin-like tonofibrils (tb) associated with the desmosomes and large aggregations of keratin-like filaments (tonofilaments; t) in the intracellular domain of the keratinocytes. Light micrograph slide was stained with hemotoxylin-eosin and magnified at (a) ×34.1; TEM: (b) ×1900, (a1) ×4800, (b1) ×6800; (a2) ×9300 and (b2) ×18 500.

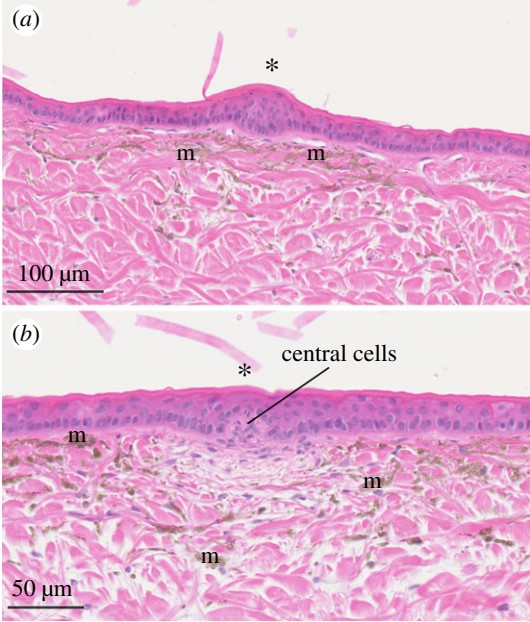

**Figure 8.** Light micrographs of transverse sections of tail skin (posterior caudal scales) of *Aipysurus laevis*. (a) Scale organs (*) in the tail are skin elevations created by a thickening of underlying epidermis. (b) Unknown dermal papillae (*) consisting of central cells that displaces surrounding *stratum germinativum* of the epidermis but does not result in skin elevations. Note that the dermis immediately underlying dermal papillae consists of loosely arranged collagen fibres devoid of melanophores (m). Slides were stained with haematoxylin-eosin and magnified at (a) ×16 and (b) ×17.2.

# 4. Discussion

## 4.1. Cephalic scale organs

### 4.1.1. Scale organs and dermal papillae

Previous work found that the cephalic scale organs of sea snakes are substantially more protruding and often cover a larger portion of the scale surface than the scale organs of terrestrial hydrophiine snakes [7]. The present study shows that, despite these differences, scale organs in sea snakes have retained a similar underlying ultrastructure to their terrestrial counterparts. The scale organs examined in *A. laevis* and *H. stokesii* are characterized by a dermal papilla that consists of an aggregation of central cells with collagen fibres, blood vessels and nerve axons in the intercellular domain that together displace the surrounding epidermis (figures 2–7). A similar underlying structure has been reported for the cephalic scale organs of terrestrial species representing several major groups of squamates including agamids, colubroids, henophidians, iguanids, scolecophidians and varanids [10,21–23,43–46]. In terrestrial snakes and sea snakes, the epidermis above the dermal papilla comprises columnar keratinocytes (i.e. *stratum germinavatum*) that form a layer that is 15% to 50% thinner than the epidermis of adjacent flat skin. The columnar keratinocytes above the dermal papilla have been described as 'cap cells' in snakes and suggested to provide protection against abrasion or aid in transducing mechanosensory stimuli [22].

We discovered that sea snake skin contained nerve axons that extend from the dermis and terminate within the alpha layer (epidermis) as distinct discoid structures (figure 5). In terrestrial colubroid snakes, these structures have variously been described as 'discoid receptors' [21], 'end bulbs' [20] and 'button-like' [10] nerve endings. In the sea snake skin, we found discoid receptors were distributed throughout the epidermis, but aggregated above the dermal papillae (figure 5a,b) deriving from axons at the base of the dermal papilla (figure 5c). This adds evidence for a sensory function of scale organs in sea snakes.

Our images from transmission electron microscopy provide the first high-resolution ultrastructure data of a cephalic scale organ in a snake. Inspection of figure 7b shows that the central cells within dermal papilla are clearly differentiated from surrounding keratinocytes by their lack of tonofilaments. Tonofilaments are formations of keratin-like proteins that provide structural integrity to the squamate epidermis [47]. Although lacking in tonofilaments, central cells maintain contact elements with surrounding keratinocytes via multiple tight junctions (desmosomes) (figure 7b, inset two). A putative axon was also identified in the intercellular domain of the dermal papilla (figure 7b, inset one), which may represent the 'terminal receptors' or myelinated axons previously identified in lizards [46]. We did not find synaptic contacts between axons and central cells, which is consistent with light microscopy studies of other colubroid snakes (e.g. *Elaphe*) [23]. Nevertheless, the presence of discoid receptors superior to the dermal papilla suggest that the central cells have a functional role in transducing mechanical stimuli.

In addition to dermal papillae associated with cephalic scale organs, we detected papillae typically (but not always) located at the base of depressions in the outer skin in *H. stokesii* (figure 4). These dermal papillae consisted of approximately 10 central cells (figure 5) and displaced surrounding keratinocytes but, in contrast to the ultrastructure we describe for cephalic scale organs, did not result in a skin elevation (bump). Putative nerve structures leading to the dermal papilla were identified using Gomori's one-step trichrome stain under light microscopy (figure 4b), but we were unable to conduct antibody staining of neuronal markers. It is unclear whether these dermal papillae are distinct scale structures or merely undeveloped or damaged scale organs.

### 4.1.2. Lamellar corpuscles

We detected lamellated, ovoid cells in the deeper dermis of cephalic skin in both species examined and demonstrated that these lamellar corpuscles were neuronal-positive in *H. stokesii* (figure 6). These structures resemble the 'non-encapsulated lamellated receptors' identified in other squamates such as *Boa*, *Elaphe*, *Iguana* and *Agama* [11,12,21,48]. The location and shape of these receptors suggest that they are small, Pacinian-like corpuscles [24,49]. Pacinian (Vater-Pacini) corpuscles are rapidly adapting low-threshold mechanoreceptors (LTMRs) that are sensitive to skin indentation and vibratory (deep touch) stimuli of high frequencies (peak 250 Hz, range 40–800 Hz), and they are present in glabrous skin of mammals [24]. Pacinian corpuscles consist of connective tissue and fibroblasts lined by flat neuronal 'Schwann' cells; the lamellar structures identified in sea snakes tested immuno-positive for the neuronal marker PgP9.5 suggesting that these are indeed modified neuronal cells. The sensitivity of these receptors has not been examined in previous electrophysiological tests of snake skin.

### 4.1.3. Ancestral and derived sensory functions for cephalic scale organs

The ultrastructural features described above for cephalic scale organs of terrestrial and marine snakes represent all of the components of Meissner-like corpuscles. Meissner corpuscles are rapidly adapting LTMRs present in the dermal papillae of mammal glabrous skin [49]. Electrophysiological experiments of cranial nerves in colubroid snakes found that they are rapidly adapting LTMRs with receptive fields that overlap with Meissner corpuscles (i.e. 12 mm$^2$, [17]). Our finding that the cephalic scale organs of sea snakes share a very similar ultrastructure with the scale organs of their terrestrial relatives (and appear to lack novel or specialized cell types) provides evidence that marine lineages have retained the ancestral mechanosensory role for these organs.

The dome shape and often high scale-coverage of mechanoreceptors in sea snakes suggests divergent selection on these organs in marine environments, either for retained (ancestral) sensitivity to tactile stimuli or a derived sensitivity to hydrodynamic stimuli. Sea snakes forage in benthic habitats, frequently probing burrows and crevices as do terrestrial snakes on land [50], but there is no obvious reason why sea snakes should require a heightened tactile sense compared to terrestrial species. It seems more likely that sea snakes have experienced selection pressures for sensitivity to hydrodynamic stimuli [7]. Observations of the sea snake *Hydrophis* (*Pelamis*) *platurus* approaching and biting a vibrating object [51] provides some behavioural evidence that sea snakes are responsive to hydrodynamic stimuli. Evoked potentials have been recorded from the midbrain of the sea snake *Hydrophis* (*Lapemis*) *curtus* in response to a vibrating sphere (50–200 Hz, peak sensitivity at 100 Hz), but no nervous response was successfully recorded directly from a scale organ [52]. However, more recently, auditory evoked potentials were recorded from the midbrain of *A. laevis* and *H. stokesii* in response to tone bursts from 40 to 600 Hz (peak sensitivity at 60 Hz) [53]. This work showed that some species of sea snakes are capable of detecting low amplitude water motion, pressure and/or particle motion. These studies were not able to discern whether water motion was detected by mechanoreceptive scale organs in the skin or hair-cells in the inner ear. However, the peak sensitivities to mechanical stimuli broadly overlap with peak sensitivities of Meissner (10–50 Hz) and Pacinian (200–300 Hz) corpuscles.

Hydrodynamic reception allows the detection of water motion, usually caused by water disturbances or animal movement, and is characterized by very low-frequency components that peak at 10 Hz with a maximum of 50 Hz. This sensory ability has evolved repeatedly in aquatic organisms wherein hydrodynamic stimuli are transduced by cutaneous mechanoreceptors (e.g. the lateral line systems in fish, cephalopods and amphibians [54,55]). Cutaneous mechanoreceptors have also been co-opted for hydrodynamic reception in secondarily aquatic mammals, a well-studied example of which are the vibrissae (whiskers) of pinnipeds [56–58].

Among snakes, two independently aquatic taxa (that are distantly related to hydrophiines) have evolved highly derived scale mechanoreceptors that putatively function to sense the water motions generated by the movement of prey. Tentacled snakes (*Erpeton tentaculum*) have the largest mechanoreceptors among vertebrates with two cephalic tentacles measuring 2–3 mm and made up of dermis, epidermis and free nerve endings [34,59]. Scale 'sensillae' in file snakes (*Acrochordus*) are thought to be sensitive to the hydrodynamic motion generated by the movement of fish prey. These small organs are vascularized like sea snake scale organs but instead of a dermal papilla, they consist of specialized epidermal cells that underlie highly derived bristles that protrude from the skin [5,15]. *Erpeton* and *Acrochordus* represent older aquatic transitions in snake phylogeny, and their mechanoreception is linked to specialized strategies for ambushing fish prey in turbid freshwater habitats of low visibility [60]. By contrast, sea snakes have recent marine origins and are a very ecologically diverse clade comprising species that variably occupy blue water reefs or turbid inshore habitats, are diurnal or nocturnal, and specialize on active or sedentary prey. The turtle-headed sea snake, *Emydocephalus annulatus*, is notable in having the second highest scale coverage of scale organs (3.8%) while being diurnally active and specializing on sessile fish eggs in clear water reefs [7,61]. These results show a complex evolutionary history of scale organs and further research should aim to link selection pressures on hydrodynamic reception with particular ecologies (e.g. optimal foraging strategies, water turbidity) among sea snakes.

## 4.2. Tail scale organs

Based on cellular morphology, there is a clear distinction between cephalic and posteriorly located scale organs in sea snakes. Scale organs present on the tail skin of *A. laevis* do not contain dermal papillae; skin elevations are instead created by a thickening of the epidermis (figure 8*a*). These structurally 'simplified' scale organs have been reported in the body skin of the sea snake *E. annulatus* [37] and the tail skin of

some terrestrial snakes [10]. Many functional roles have been proposed for body scale organs in sea snakes, including mechanoreception, mate recognition, and enhanced friction for improved swimming performance, gripping and/or ecdysis [7,26,37]. We were unable to stain for the presence of free nerve endings in tail scale organs; however, nerve staining of 'supracloacal tubercles' in the snakes *Thamnophis sirtalis* and *Nerodia rhombifer* (formerly *Natrix rhombifera*) found that they were innervated in a similar pattern to cephalic scale organs, and thought to be important for sensory feedback to aid alignment of the cloacae during copulation [10,62]. Although posteriorly located scale organs exhibit clear ultrastructural differences compared to cephalic scale organs, it is likely that they have mechanoreceptive and/or structural functions in sea snakes.

The ultrastructural differences in cephalic scale organs versus posteriorly located scale organs may reflect variation in mechanosensitivity in the head compared to the rest of the body. Research in mammals suggests that the structure of the skin organ may be just as important as the neurons that carry the electrical impulse; collagen can provide physical tethering, structural integrity, or aid in propagating or modulating the sensation of force [25]. The absence of a dermal papilla for scale organs on the body and tail skin might indicate a less specialized mechanoreceptor with differential sensitivity compared to a cephalic mechanoreceptor. Thus, our results support previous studies on terrestrial snakes that suggest that the head of sea snakes is the prime exploratory organ for mechanical stimulation [17,27]. Future studies should investigate the neural pathways and compare electrophysiological responses underlying scale mechanoreceptors distributed on the head and body of snakes. Such efforts may discover that sea snakes possess specialized nerve pathways and/or responsive fields that are analogous to the cranial nerve canals of neuromasts in fish and amphibians, or the vibrissae of secondarily aquatic systems in mammals [57,63], which would support a hydrodynamic function for cephalic scale organs.

## 4.3. Dermal photoreception and other cutaneous sensory modalities

The skin provides a primary interface for receiving multiple stimuli, creating an opportunity for multimodal cutaneous receptors. Indeed, molecular and electrophysiological studies of ISOs in crocodiles indicate multi-modal sensitivity to mechanical stimuli, and thermal and pH gradients [64,65]. Dermal photoreceptors in the tail skin of *Aipysurus* sea snakes mediate phototactic behaviour in these species [36]: we did not detect candidate photoreceptive structures (e.g. photoreceptors, stacked membranes) in the tail skin of *A. laevis*, but we did find structurally simplified scale organs (described above) and other small dermal papilla (figure 8*b*). Given that cutaneous receptors have been linked with both mechano- and photoreception in amphibians [66] and marine invertebrates [67], these scale organs merit further investigation for their putative role in photoreception.

Several other sensory functions have been tentatively attributed to the scale organs of sea snakes, but these currently lack supporting evidence. An electro-magneto-sense is plausible [5], but our histological sections do not show canals or pores that are indicative of passive electroreceptors (e.g. ampullary-type organs) or specialized active electroreceptive organs (e.g. tuberous organs or mormyromasts of weakly electric fish) [68,69]. Similarly, in addition to previous studies using scanning electron microscopy [5,7], our study demonstrates that scale organs in sea snakes are devoid of pores and so a chemosensory function is highly unlikely. Baroreception of the changes in air pressure that precede extreme weather events has been attributed to sea kraits [70], which are an independently marine clade of hydrophiines; however, it is unclear whether sea snakes react in a similar way and how cutaneous mechanoreceptors might transduce pressure information in sea snakes or sea kraits. Salinity is an important predictor of sea snake distribution [71] because many species require access to freshwater for hydration [72–74], but pH receptors are more likely to be located in papillae in the mouth [11,75]. The thermal sensitivity of scale organs has been investigated in colubroid (*Elaphe*) snakes with results indicating that although some cutaneous nerves are exclusively sensitive to heat, mechanoreceptive fibres are not responsive to either heating or cooling [17,76]. Finally, these sensory hypotheses do not exclude other non-sensory functions for scale organs, e.g. modifying boundary layer of skin, so these roles should be considered in future studies of the evolution of the scale organs in sea snakes.

## 5. Conclusion

Our study shows that the ultrastructure of cephalic scale organs of sea snakes closely resembles the mechanosensitive Meissner-like corpuscles that underlie the scale organs of terrestrial snakes. This

provides evidence that the scale organs of marine hydrophiines have retained an ancestral mechanosensory function. Our findings provide the basis for future research into the sensitivity of cutaneous receptors in sea snakes including mechano-, hydro- and photo-sensory modalities. Our study highlights that snakes are an important group for understanding the evolution of mechanoreception in vertebrates, particularly in response to shifting sensory landscapes.

Ethics. Animals were collected in accordance with the Western Australian Department of Biodiversity, Conservation and Attractions (SF010002). Euthanasia was carried out in accordance with the guidelines of the Australian Code of Practice for the Care and Use of Animals for Scientific Purposes, under Animal Ethics Committee protocols from the University of Adelaide (S-2015-119) and the University of Western Australia (RA/3/100/1369).

Data accessibility. Electronic supplementary data are interactive digital scans of slides (ndp.view files) and skin measurements (.xlxs) for *A. laevis* and *H. stokesii*. These are available with the electronic supplementary material at Figshare, doi:10.25909/5c5bb6777f249.

Authors' contributions. J.M.C.-R. and K.L.S. conceived of the study. J.M.C.-R., L.C. and K.L.S collected samples; J.M.C.-R. and R.W. carried out microscopy analysis and interpretation with input from L.C.; J.M.C.-R. and K.L.S wrote the manuscript with input from L.C. and R.W.

Competing interests. The authors declare no competing interests.

Funding. This work was supported by a Hermon-Slade Foundation Grant (0001039517) and Future Fellowship to K.L.S. (FT130101965), and an Australian Government Research Training Program Scholarship and Fulbright Postgraduate Scholarship held by J.M.C-R.

Acknowledgements. We are grateful to Kylie Sherwood (Chelonia Broome), Caroline Kerr (The University of Western Australia), Mick and Kelly Woodley and crew (Absolute Ocean Charters, Broome) for assistance in collecting sea snakes. We thank Luke Allen (Venom Supplies Pty Ltd, South Australia) for supplying taipan tissue. For access to specimens and laboratories at the South Australian Museum, we thank Mark Hutchinson and Carolyn Kovach. We thank Kathryn Batra, Chris Leigh and Jim Manavis (Adelaide Medical School, University of Adelaide), Peter Hill and Lucy Woolford (School of Animal and Veterinary Sciences, University of Adelaide), and Jane Sibbons (Adelaide Microscopy, South Australia) for assistance with immunohistochemistry and microscopy analyses and interpretation. We are grateful to the anonymous reviewers for their comments that improved this manuscript.

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
