## [Reviewer comments · Royal Society Open Science]

Review History

RSOS-182022.R0 (Original submission)

Review form: Reviewer 1 (Duncan Leitch)

Is the manuscript scientifically sound in its present form?

Yes

Are the interpretations and conclusions justified by the results?

Yes

Is the language acceptable?

Yes

Is it clear how to access all supporting data?

Yes

Do you have any ethical concerns with this paper?

No

Have you any concerns about statistical analyses in this paper?

No

Recommendation?

Accept with minor revision (please list in comments)

Comments to the Author(s)

The paper provides compelling anatomical evidence regarding the specialized mechanosensory function of cephalic and tail scale organs from two species of sea snake. I appreciate that the authors have used a variety of histological and EM techniques to describe the unique morphology of these organs, including careful measure of adjacent skin layers, suggesting specialized tactile function, similar to the mechanosensory organs noted in terrestrial snakes.

I have a few suggestions:

- 1) A summary schematic figure would serve this paper well. I suggest two (or more panels) representing the specialized cephalic organs, as well as the distinct tail organs. This could be prepared from observations from all the microscopic techniques in total (e.g., relative location of putative neuronal bundles from the PGP immunohistochemistry, numbers of center cells as based on the counts from TEM, etc.) and would serve the authors well in visually showing the differences in these organs, such as locations/numbers of discoid organs, lamellated corpuscles, differences in thickness of the skin strata.
- 2) The paper could benefit with a little more discussion regarding the potential innervation of these organs. The work of comparative neuroanatomists like Glenn Northcutt have showed elaborate branching patterns that suggest particular sensory function/importance in fish, reptile, and amphibian species, and these patterns have often been used to make evolutionary inferences. The authors touch upon this in the "Tail scale organs" section in the discussion but this can be taken a bit further. Innervation to the hydrodynamic sensors of the lateral line system- the neuromasts - arise through their own specialized cranial nerve system rather than trigeminal.
- 3) The authors could communicate with a little more confidence about the unlikely role of these sense organs in electroreception (as in the last sentence of the Dermal photoreception... section). Their histological sections do not seem to show any kind of canal or pore typical of passive electroreceptors, like ampullary-type organs. Also, based on the sections shown in the paper, it doesn't appear that their are more specialized active electroreceptive organs, like tuberous organs or mormyromasts seen in freshwater weakly-electric fish.

Other minor correction/comments:

- 1) The background white balancing looks unusual among panels in Figs. 5 and 6.
- 2) The labels are difficult to read in Fig. 7, with the black labeling on top of the black and white TEM. Perhaps the brightness or contrast could be adjusted, or the labels themselves colored.
- 3) The putative myelinated axon looks unusual with the sheathing appear very thin. The authors acknowledge that this is the best resolution available, however.
- 4) In paragraph 2 of the discussion, the authors state that they can see free nerve axons terminating in the alpha layers in Fig. 3. Free nerve endings are a specific kind of specialized sensory end organ, often described as related to pain sensation. The nerve bundles adjacent to the putative sense organs are visible, but free nerve endings terminating in the the distal epidermis don't seem visible here (maybe just at this resolution).

Review form: Reviewer 2 (Kurt Schwenk)

Is the manuscript scientifically sound in its present form?

Yes

Are the interpretations and conclusions justified by the results?

Yes

Is the language acceptable?

Yes

Is it clear how to access all supporting data?

Yes

Do you have any ethical concerns with this paper?

No

Have you any concerns about statistical analyses in this paper?

No

Recommendation?

Accept with minor revision (please list in comments)

Comments to the Author(s)

In a previous study the authors found that fully marine, hydrophiine snakes had, on average, a higher density of scale organs on cranial scales than their terrestrial relatives. Scale organs also tended to be larger and cover a larger area of the scale, although there was broad overlap in these measures. In the present study the authors consider whether the derived scale organs of marine species have retained the ancestral mechanoreceptive (terrestrial) function, or if they have also changed in function. They use light and transmission electron microscopy, and immunohistochemistry to identify neuronal tissue to address the question. They show convincingly that the scale receptors retain a mechanoreceptive (touch) function.

The study is sound, the conclusions reasonable and supported by the data. I have only a few general comments and several small, editorial comments.

GENERAL COMMENTS

(1) [p. 4, 1st paragraph; p. 11, lines 335-336; p. 12, lines 339-340] There is some ambiguity/mischaracterization about the role of mechanoreceptors in sensing 'touch' vs. "hydrodynamic stimuli." I believe that this is a false distinction. The evidence is that the snake scale organs are mechanoreceptors sensitive to pressure. Pressure can take the form of physical touch or a compression wave traveling through water (a hydrodynamic stimulus). Thus there is not a dichotomy between 'touch receptors' and 'hydrodynamic receptors'. The evidence suggests, circumstantially, that the scale organs of snakes are adapted to be more sensitive to pressure/mechanical stimuli than their terrestrial counterparts, presumably because water pressure waves have less energy than a physical touch. To be clear, I believe that this is what the authors mean to say, but as written, it is either unclear or the distinction is overstated. The distinction relates to the nature or source of mechanical, pressure stimuli, not (necessarily) to the nature of the receptors.

Related to the above, note that the Merkel cell neurite complexes characteristic of crocodylian ISOs are also mechanosensory and hence would not actually play an "alternative sensory role"

(p. 4, lines 94-95). Finally, it is suggested (p. 4, lines 80-82) that *Acrochordus* uses its scale organs to sense “water motion.” I do not believe this is true. My understanding is that the snakes respond to physical touch of the fish. This should be checked.

(2) Throughout the paper the scale organs are referred to as “sensilla”. I understand that this term is often used in the literature to refer to similar structures in squamate reptiles. However, it is inaccurate and misleading as a sensillum is a hair-like structure (usually in insects or other invertebrate taxa). It is also true that many lizard scale organs have a central hair-like protrusion that could more accurately be called a sensillum (exclusive of the rest of the receptor). Other terms are used in the literature for the sensory organs described in this paper, including ‘scale organ’ and ‘integumentary sensory organ’ or ‘ISO’. These would seem to me to be far preferable than “sensilla/sensillum”, but obviously this is the choice of the authors.

(3) Throughout the description of the sensory organs its inner, dermal component is referred to as a “dermal capsule”. In my opinion, this is inaccurate and inconsistent with most anatomical usage. A capsule represents a discrete covering or sheath that surrounds something [“a membranous structure...that envelopes an organ” – *Stedman’s Medical Dictionary*, 27th ed.], which is not the case here. The dermal protrusion into the epidermis is more accurately described as a ‘dermal papilla’, as parenthetically noted on p. 5, line 134. This should be the term used throughout the paper.

(4) Note that a finding of tactile/mechanoreception does not exclude all other possible functions, e.g., they could still function in modifying flow over the snake’s surface (though unlikely).

EDITORIAL COMMENTS

Line 40: I do not believe that there is any evidence to support the assertion that snakes use mechanoreceptors to discriminate prey types. This is pure speculation based on mixed receptor types within the mouth. Evidence suggests that virtually all prey discrimination is chemosensory (gustatory and vomeronasal).

Line 45: change “stimulus” to ‘stimuli’

Line 95: delete hyphen in “Merkel-cell”

Line 104: “collected 1 10 km offshore”; seems to be a typo, not clear what it should be

Line 127: and elsewhere; insert ‘trichrome after “Gomori’s one-step”; also, do not capitalize “One-Step”

Line 130: delete “the height (thickness)” and change to ‘thickness’

Line 184: It is unclear what is meant by “horizontally arranged” in reference to the ‘central cells’; in the images they are either clustered, vertical or circular – never horizontal

Line 187: re: the apparently basal taper in *H. stokesii* – are you sure that this is not simply a plane of section issue? Did you have serial sections across the width of the receptors to confirm this? From other images it appears that the dermal papilla extends outward/laterally toward the scale surface, i.e., the distal part of the papilla is wider than its base. As such, a section that just passes through the point that a lateral extension joins the central core would look tapered toward the base.

Line 204: change “skin” to ‘scale’ for clarity

Line 238: change “present at base of dermal” to ‘present at the base of the dermal’

Line 247: replace the comma with a period; start new sentence with “The outer bumps...”

Line 279: The comment about “cap cells” based on Jackson (1977) seems a bit pointless. There is no histological difference between the keratinocytes covering the dermal papilla and others. Obviously any cells in this position would provide abrasion resistance, but no moreso than anywhere else.

Lines 286-287: The figures do not show any direct evidence of discoid receptor innervation that I can see. Obviously they must be innervated and the nerves get pretty close within the scale organ, but no nerves leading directly to the discoid receptors, particularly the more distant, epidermal receptors, are evident.

Lines 296-299: It seems implausible to me that the central cells have no sensory function. What’s the point of the whole organ structure then, particularly given that discoid receptors are distributed all over? How confident are you about this, i.e, what is the probability that you would have seen synaptic complexes? There are neurons all over within the dermal papilla and it seems unlikely that they are merely supplying discoid receptors. What else would they be doing? Just free nerve endings?

Lines 359-362: Catania (1995), ref. 59, does not provide any evidence for sensitivity to hydrodynamic stimuli in star-nosed moles, nor can I find any other reference to such a thing. They definitely use the star organ for direct touch of food objects while foraging within water, but again (see General Comment 1), this is no different from terrestrial touch. I did not check ref. 60 for the platypus, but I would confirm that they do, indeed, have receptors that are sensitive to water movement AND that they have been “co-opted” from terrestrial cutaneous touch receptors. In fact, I would confirm this for all of them. Direct touch underwater is not the same thing as being used to detect water movement (hydrodynamic stimuli). Pinniped whiskers are a good example for mammals (they use them to detect vortices indicating fish trails) [see review by Dehnhardt & Mauck (2008), pp. 295-314, In *Sensory Evolution on the Threshold*, JGM Thewissen and S. Nummela (eds.), Univ. of California Press, Berkeley, CA]. also manatee whiskers and body hairs [reviewed in Bauer et al. (2018) *The tactile senses of marine mammals*. *Int. J. Comp. Psychology* 31, special issue, M. Botero ed.].

Line 401: insert ‘compared to a’ after “differential sensitivity”

Line 402: I don’t see why the trigeminal or other cranial nerves that innervate cephalic cutaneous receptors are “specialized” ... The sensory organs they innervate might be specialized receptors, but the cranial nerves, themselves, are not specialized.

Line 404: the “dorsal root ganglion” is not a “peripheral nerve of the spinal cord” –it is a part of the spinal nerve within which the sensory nerve bodies lie. For lines 402-404, it is sufficient to note that the cephalic receptors are innervated by cranial nerves while the postcranial receptors are innervated by spinal nerves, which is exactly what one would expect.

Line 406: the receptors are not used “to actively seek” – they are used while he snake actively seeks...

Line 409: do you mean ‘electrophysiological’ rather than ‘electrophysical’?

Lines 429-432: there is also no histological support for a magnetic sense

FIGURE LEGENDS: 2A: no plane of section is give for the image; 2B: delete “cross-” (redundant); ‘hematoxylin’ is misspelled “hemotoxylin”; 2C: insert ‘trichrome’ after “Gomori one-step”; 3A: no plane of section given; 3B,C: delete “cross-”; ‘transverse’ is misspelled “traverse”; 4: check for above changes; also, Latin binomial name not italicized; “deeper cross-section” doesn’t make sense to me, please clarify; 7: “keratin filaments tonofilaments (t)” must be a typo – missing parentheses?

Decision letter (RSOS-182022.R0)

15-Feb-2019

Dear Ms Crowe-Riddell

On behalf of the Editors, I am pleased to inform you that your Manuscript RSOS-182022 entitled "Ultrastructural evidence of a mechanosensory function of scale ‘sensilla’ in sea snakes (Hydrophiinae)" has been accepted for publication in Royal Society Open Science subject to minor revision in accordance with the referee suggestions. Please find the referees' comments at the end of this email.

The reviewers and handling editors have recommended publication, but also suggest some minor revisions to your manuscript. Therefore, I invite you to respond to the comments and revise your manuscript.

- Ethics statement

- Data accessibility

<http://datadryad.org/submit?journalID=RSOS&manu=RSOS-182022>

- Competing interests

- Authors’ contributions

All submissions, other than those with a single author, must include an Authors’ Contributions section which individually lists the specific contribution of each author. The list of Authors

should meet all of the following criteria; 1) substantial contributions to conception and design, or acquisition of data, or analysis and interpretation of data; 2) drafting the article or revising it critically for important intellectual content; and 3) final approval of the version to be published.

- Acknowledgements

- Funding statement

Because the schedule for publication is very tight, it is a condition of publication that you submit the revised version of your manuscript before 24-Feb-2019. Please note that the revision deadline will expire at 00.00am on this date. If you do not think you will be able to meet this date please let me know immediately.

- 1) A text file of the manuscript (tex, txt, rtf, docx or doc), references, tables (including captions) and figure captions. Do not upload a PDF as your "Main Document";

- 2) A separate electronic file of each figure (EPS or print-quality PDF preferred (either format should be produced directly from original creation package), or original software format);
- 3) Included a 100 word media summary of your paper when requested at submission. Please ensure you have entered correct contact details (email, institution and telephone) in your user account;
- 4) Included the raw data to support the claims made in your paper. You can either include your data as electronic supplementary material or upload to a repository and include the relevant doi within your manuscript. Make sure it is clear in your data accessibility statement how the data can be accessed;
- 5) All supplementary materials accompanying an accepted article will be treated as in their final form. Note that the Royal Society will neither edit nor typeset supplementary material and it will be hosted as provided. Please ensure that the supplementary material includes the paper details where possible (authors, article title, journal name).

on behalf of Dr Richard Benton (Associate Editor) and Kevin Padian (Subject Editor)
openscience@royalsociety.org

Reviewer comments to Author:
Reviewer: 1

Comments to the Author(s)
The paper provides compelling anatomical evidence regarding the specialized mechanosensory function of cephalic and tail scale organs from two species of sea snake. I appreciate that the

authors have used a variety of histological and EM techniques to describe the unique morphology of these organs, including careful measure of adjacent skin layers, suggesting specialized tactile function, similar to the mechanosensory organs noted in terrestrial snakes.

I have a few suggestions:

1) A summary schematic figure would serve this paper well. I suggest two (or more panels) representing the specialized cephalic organs, as well as the distinct tail organs. This could be prepared from observations from all the microscopic techniques in total (e.g., relative location of putative neuronal bundles from the PGP immunohistochemistry, numbers of center cells as based on the counts from TEM, etc.) and would serve the authors well in visually showing the differences in these organs, such as locations/numbers of discoid organs, lamellated corpuscles, differences in thickness of the skin strata.

2) The paper could benefit with a little more discussion regarding the potential innervation of these organs. The work of comparative neuroanatomists like Glenn Northcutt have showed elaborate branching patterns that suggest particular sensory function/importance in fish, reptile, and amphibian species, and these patterns have often been used to make evolutionary inferences. The authors touch upon this in the "Tail scale organs" section in the discussion but this can be taken a bit further. Innervation to the hydrodynamic sensors of the lateral line system- the neuromasts - arise through their own specialized cranial nerve system rather than trigeminal.

3) The authors could communicate with a little more confidence about the unlikely role of these sense organs in electroreception (as in the last sentence of the Dermal photoreception... section). Their histological sections do not seem to show any kind of canal or pore typical of passive electroreceptors, like ampullary-type organs. Also, based on the sections shown in the paper, it doesn't appear that their are more specialized active electroreceptive organs, like tuberous organs or mormyromasts seen in freshwater weakly-electric fish.

Other minor correction/comments:

1) The background white balancing looks unusual among panels in Figs. 5 and 6.

2) The labels are difficult to read in Fig. 7, with the black labeling on top of the black and white TEM. Perhaps the brightness or contrast could be adjusted, or the labels themselves colored.

3) The putative myelinated axon looks unusual with the sheathing appear very thin. The authors acknowledge that this is the best resolution available, however.

4) In paragraph 2 of the discussion, the authors state that they can see free nerve axons terminating in the alpha layers in Fig. 3. Free nerve endings are a specific kind of specialized sensory end organ, often described as related to pain sensation. The nerve bundles adjacent to the putative sense organs are visible, but free nerve endings terminating in the the distal epidermis don't seem visible here (maybe just at this resolution).

Reviewer: 2

Comments to the Author(s)

In a previous study the authors found that fully marine, hydrophiine snakes had, on average, a higher density of scale organs on cranial scales than their terrestrial relatives. Scale organs also tended to be larger and cover a larger area of the scale, although there was broad overlap in these measures. In the present study the authors consider whether the derived scale organs of marine species have retained the ancestral mechanoreceptive (terrestrial) function, or if they have also changed in function. They use light and transmission electron microscopy, and immunohistochemistry to identify neuronal tissue to address the question. They show convincingly that the scale receptors retain a mechanoreceptive (touch) function.

The study is sound, the conclusions reasonable and supported by the data. I have only a few general comments and several small, editorial comments.

GENERAL COMMENTS

(1) [p. 4, 1st paragraph; p. 11, lines 335-336; p. 12, lines 339-340] There is some ambiguity/mischaracterization about the role of mechanoreceptors in sensing 'touch' vs. "hydrodynamic stimuli." I believe that this is a false distinction. The evidence is that the snake scale organs are mechanoreceptors sensitive to pressure. Pressure can take the form of physical touch or a compression wave traveling through water (a hydrodynamic stimulus). Thus there is not a dichotomy between 'touch receptors' and 'hydrodynamic receptors'. The evidence suggests, circumstantially, that the scale organs of snakes are adapted to be more sensitive to pressure/mechanical stimuli than their terrestrial counterparts, presumably because water pressure waves have less energy than a physical touch. To be clear, I believe that this is what the authors mean to say, but as written, it is either unclear or the distinction is overstated. The distinction relates to the nature or source of mechanical, pressure stimuli, not (necessarily) to the nature of the receptors.

Related to the above, note that the Merkel cell neurite complexes characteristic of crocodylian ISOs are also mechanosensory and hence would not actually play an "alternative sensory role" (p. 4, lines 94-95). Finally, it is suggested (p. 4, lines 80-82) that *Acrochordus* uses its scale organs to sense "water motion." I do not believe this is true. My understanding is that the snakes respond to physical touch of the fish. This should be checked.

(2) Throughout the paper the scale organs are referred to as "sensilla". I understand that this term is often used in the literature to refer to similar structures in squamate reptiles. However, it is inaccurate and misleading as a sensillum is a hair-like structure (usually in insects or other invertebrate taxa). It is also true that many lizard scale organs have a central hair-like protrusion that could more accurately be called a sensillum (exclusive of the rest of the receptor). Other terms are used in the literature for the sensory organs described in this paper, including 'scale organ' and 'integumentary sensory organ' or 'ISO'. These would seem to me to be far preferable than "sensilla/sensillum", but obviously this is the choice of the authors.

(3) Throughout the description of the sensory organs its inner, dermal component is referred to as a "dermal capsule". In my opinion, this is inaccurate and inconsistent with most anatomical usage. A capsule represents a discrete covering or sheath that surrounds something ["a membranous structure...that envelops an organ" – *Stedman's Medical Dictionary*, 27th ed.], which is not the case here. The dermal protrusion into the epidermis is more accurately described as a 'dermal papilla', as parenthetically noted on p. 5, line 134. This should be the term used throughout the paper.

(4) Note that a finding of tactile/mechanoreception does not exclude all other possible functions, e.g., they could still function in modifying flow over the snake's surface (though unlikely).

EDITORIAL COMMENTS

Line 40: I do not believe that there is any evidence to support the assertion that snakes use mechanoreceptors to discriminate prey types. This is pure speculation based on mixed receptor types within the mouth. Evidence suggests that virtually all prey discrimination is chemosensory (gustatory and vomeronasal).

Line 45: change "stimulus" to 'stimuli'

Line 95: delete hyphen in “Merkel-cell”

Line 104: “collected 1 10 km offshore”; seems to be a typo, not clear what it should be

Line 127: and elsewhere; insert ‘trichrome after “Gomori’s one-step”; also, do not capitalize “One-Step”

Line 130: delete “the height (thickness)” and change to ‘thickness’

Line 184: It is unclear what is meant by “horizontally arranged” in reference to the ‘central cells’; in the images they are either clustered, vertical or circular – never horizontal

Line 187: re: the apparently basal taper in *H. stokesii* – are you sure that this is not simply a plane of section issue? Did you have serial sections across the width of the receptors to confirm this? From other images it appears that the dermal papilla extends outward/laterally toward the scale surface, i.e., the distal part of the papilla is wider than its base. As such, a section that just passes through the point that a lateral extension joins the central core would look tapered toward the base.

Line 204: change “skin” to ‘scale’ for clarity

Line 238: change “present at base of dermal” to ‘present at the base of the dermal’

Line 247: replace the comma with a period; start new sentence with “The outer bumps...”

Line 279: The comment about “cap cells” based on Jackson (1977) seems a bit pointless. There is no histological difference between the keratinocytes covering the dermal papilla and others. Obviously any cells in this position would provide abrasion resistance, but no more so than anywhere else.

Lines 286-287: The figures do not show any direct evidence of discoid receptor innervation that I can see. Obviously they must be innervated and the nerves get pretty close within the scale organ, but no nerves leading directly to the discoid receptors, particularly the more distant, epidermal receptors, are evident.

Lines 296-299: It seems implausible to me that the central cells have no sensory function. What’s the point of the whole organ structure then, particularly given that discoid receptors are distributed all over? How confident are you about this, i.e, what is the probability that you would have seen synaptic complexes? There are neurons all over within the dermal papilla and it seems unlikely that they are merely supplying discoid receptors. What else would they be doing? Just free nerve endings?

Lines 359-362: Catania (1995), ref. 59, does not provide any evidence for sensitivity to hydrodynamic stimuli in star-nosed moles, nor can I find any other reference to such a thing. They definitely use the star organ for direct touch of food objects while foraging within water, but again (see General Comment 1), this is no different from terrestrial touch. I did not check ref. 60 for the platypus, but I would confirm that they do, indeed, have receptors that are sensitive to water movement AND that they have been “co-opted” from terrestrial cutaneous touch receptors. In fact, I would confirm this for all of them. Direct touch underwater is not the same thing as being used to detect water movement (hydrodynamic stimuli). Pinniped whiskers are a good example for mammals (they use them to detect vortices indicating fish trails) [see review by Dehnhardt & Mauck (2008), pp. 295-314, In *Sensory Evolution on the Threshold*, JGM Thewissen

and S. Nummela (eds.), Univ. of California Press, Berkeley, CA]. also manatee whiskers and body hairs [reviewed in Bauer et al. (2018) The tactile senses of marine mammals. *Int. J. Comp. Psychology* 31, special issue, M. Botero ed.].

Line 401: insert 'compared to a' after "differential sensitivity"

Line 402: I don't see why the trigeminal or other cranial nerves that innervate cephalic cutaneous receptors are "specialized" ... The sensory organs they innervate might be specialized receptors, but the cranial nerves, themselves, are not specialized.

Line 404: the "dorsal root ganglion" is not a "peripheral nerve of the spinal cord" – it is a part of the spinal nerve within which the sensory nerve bodies lie. For lines 402-404, it is sufficient to note that the cephalic receptors are innervated by cranial nerves while the postcranial receptors are innervated by spinal nerves, which is exactly what one would expect.

Line 406: the receptors are not used "to actively seek" – they are used while he snake actively seeks...

Line 409: do you mean 'electrophysiological' rather than 'electrophysical'?

Lines 429-432: there is also no histological support for a magnetic sense

FIGURE LEGENDS: 2A: no plane of section is give for the image; 2B: delete "cross-" (redundant); 'hematoxylin' is misspelled "hemotoxylin"; 2C: insert 'trichrome' after "Gomori one-step"; 3A: no plane of section given; 3B,C: delete "cross-"; 'transverse' is misspelled "traverse"; 4: check for above changes; also, Latin binomial name not italicized; "deeper cross-section" doesn't make sense to me, please clarify; 7: "keratin filaments tonofilaments (t)"? must be a typo – missing parentheses?

Author's Response to Decision Letter for (RSOS-182022.R0)

See Appendices A & B.

Decision letter (RSOS-182022.R1)

15-Mar-2019

Dear Ms Crowe-Riddell,

I am pleased to inform you that your manuscript entitled "Ultrastructural evidence of a mechanosensory function of scale organs ('sensilla') in sea snakes (Hydrophiinae)" is now accepted for publication in Royal Society Open Science.

You can expect to receive a proof of your article in the near future. Please contact the editorial office (openscience_proofs@royalsociety.org and openscience@royalsociety.org) to let us know if

you are likely to be away from e-mail contact. Due to rapid publication and an extremely tight schedule, if comments are not received, your paper may experience a delay in publication.

on behalf of Dr Richard Benton (Associate Editor) and Professor Kevin Padian (Subject Editor)
openscience@royalsociety.org

Appendix A

Dear Andrew Dunn,

We are delighted to have our paper “Ultrastructural evidence of a mechanosensory function of scale ‘sensilla’ in sea snakes (Hydrophiinae)” accepted for publication in Open Science. We are grateful for the detailed and constructive comments that have enhanced the quality of the manuscript. We have made the recommended changes, all of which are minor, with one exception: unfortunately, we were not able to include a schematic diagram (as suggested by Reviewer 1) at this time. Please see our responses to reviewer comments below.

Kind regards,

Jenna Crowe-Riddell and co-authors

Reviewer comments to Author:

Reviewer: 1

Comments to the Author(s)

The paper provides compelling anatomical evidence regarding the specialized mechanosensory function of cephalic and tail scale organs from two species of sea snake. I appreciate that the authors have used a variety of histological and EM techniques to describe the unique morphology of these organs, including careful measure of adjacent skin layers, suggesting specialized tactile function, similar to the mechanosensory organs noted in terrestrial snakes.

Reviewer 1	
Reviewer 1 comments	Author's response
1) A summary schematic figure would serve this paper well. I suggest two (or more panels) representing the specialized cephalic organs, as well as the distinct tail organs. This could be prepared from observations from all the microscopic techniques in total (e.g., relative location of putative neuronal bundles from the PGP immunohistochemistry, numbers of center cells as based on the counts from TEM, etc.) and would serve the authors well in visually showing the differences in these organs, such as locations/numbers of discoid organs, lamellated corpuscles, differences in thickness of the skin strata.	This is a great suggestion and would improve the paper. Unfortunately, we have not had time to make this change.
2) The paper could benefit with a little more discussion regarding the potential innervation of these organs. The work of comparative neuroanatomists like Glenn Northcutt have showed elaborate branching patterns that suggest particular sensory function/importance in fish, reptile, and amphibian species, and these patterns have often been used to make evolutionary inferences. The authors touch upon this in the "Tail scale organs" section in the discussion but this can be taken a bit further. Innervation to the hydrodynamic sensors of the	We have incorporated relevant information from the Glenn Northcutt review paper on the comparative nerve patterns of lateral line systems in the 'Tail scale organs' section in the discussion and added the following sentence (lines 413-418): “Future studies should investigate the neural pathways and compare electrophysiological responses underlying scale mechanoreceptors distributed on the head and body of snakes. Such efforts may discover that sea snakes possess specialised nerve pathways

lateral line system- the neuromasts - arise through their own specialized cranial nerve system rather than trigeminal.	and/or responsive fields that are analogous to the cranial nerve canals of neuromasts in fish and amphibians, or the vibrissae of secondarily-aquatic systems in mammals (55,61), which would support a hydrodynamic function for cephalic scale organs.”
3) The authors could communicate with a little more confidence about the unlikely role of these sense organs in electroreception (as in the last sentence of the Dermal photoreception... section). Their histological sections do not seem to show any kind of canal or pore typical of passive electroreceptors, like ampullary-type organs. Also, based on the sections shown in the paper, it doesn't appear that their are more specialized active electroreceptive organs, like tuberous organs or mormyromasts seen in freshwater weakly-electric fish.	We have made this change to paragraph ‘Tail scale organs’ in the discussion, such that (lines 432-435): “An electro-magneto-sense is plausible (5), but our histological sections do not show canals or pores that are indicative of passive electroreceptors (e.g. ampullary-type organs) or specialised active electroreceptive organs (e.g. tuberous organs or mormyromasts of weakly-electric fish) (66,67).”
Other minor correction/comments:	
1) The background white balancing looks unusual among panels in Figs. 5 and 6.	The white balance has been corrected for Figs. 5 and 6.
2) The labels are difficult to read in Fig. 7, with the black labeling on top of the black and white TEM. Perhaps the brightness or contrast could be adjusted, or the labels themselves colored.	Font size has been increased and changed to bold. Scale bars and font have also been increased.
3) The putative myelinated axon looks unusual with the sheathing appear very thin. The authors acknowledge that this is the best resolution available, however.	Unfortunately, we were unable to achieve a higher resolution image of the putative myelinated axon.
4) In paragraph 2 of the discussion, the authors state that they can see free nerve axons terminating in the alpha layers in Fig. 3. Free nerve endings are a specific kind of specialized sensory end organ, often described as related to pain sensation. The nerve bundles adjacent to the putative sense organs are visible, but free nerve endings terminating in the the distal epidermis don't seem visible here (maybe just at this resolution).	This was a typo- meant to reference Figure 5 (immunohistochemical results), this change has made as well as removing “free” before “nerve axons”

Reviewer: 2

Comments to the Author(s)

In a previous study the authors found that fully marine, hydrophiine snakes had, on average, a higher density of scale organs on cranial scales than their terrestrial relatives. Scale organs also tended to be larger and cover a larger area of the scale, although there was broad overlap in these measures. In the present study the authors consider whether the derived scale organs of marine species have retained the ancestral mechanoreceptive (terrestrial) function, or if they have also changed in function. They use light and transmission electron microscopy, and

immunohistochemistry to identify neuronal tissue to address the question. They show convincingly that the scale receptors retain a mechanoreceptive (touch) function.

The study is sound, the conclusions reasonable and supported by the data. I have only a few general comments and several small, editorial comments.

GENERAL COMMENTS

Reviewer 2 comments	Author's response
(1) [p. 4, 1st paragraph; p. 11, lines 335-336; p. 12, lines 339-340] There is some ambiguity/mischaracterization about the role of mechanoreceptors in sensing 'touch' vs. "hydrodynamic stimuli." I believe that this is a false distinction. The evidence is that the snake scale organs are mechanoreceptors sensitive to pressure. Pressure can take the form of physical touch or a compression wave traveling through water (a hydrodynamic stimulus). Thus there is not a dichotomy between 'touch receptors' and 'hydrodynamic receptors'. The evidence suggests, circumstantially, that the scale organs of snakes are adapted to be more sensitive to pressure/mechanical stimuli than their terrestrial counterparts, presumably because water pressure waves have less energy than a physical touch. To be clear, I believe that this is what the authors mean to say, but as written, it is either unclear or the distinction is overstated. The distinction relates to the nature or source of mechanical, pressure stimuli, not (necessarily) to the nature of the receptors.	We understand the reviewer's concern and have clarified the relationship between mechanosensitivity to touch versus water motion in the Abstract (lines 19-22), Introduction (lines 76-80; 91-93) and Discussion (lines 340-346; 360-364) However, we are reluctant to use of the term 'pressure' for the following reasons. Water movement can consist of mid water 'pressure waves' or water surface waves. But the term 'pressure wave' has a complex and often confused interpretation. A vibrating object in the water, for example, can produce change in pressure (this is what we call 'sound', and we can measure it in Pascals but will also produce displacement of the particles in the medium (which is NOT pressure). To detect the pressure wave (in Pascal), you need a pressure-sensitive device (pressure - displacement transducer). Pressure detection is mediated by the inner ears of land vertebrates (which can be considered mechanoreceptors), and by the swim bladders of some fishes. It is possible that the inner ears of sea snakes also detect pressure, but this remains to be proven. Based on this, we consider it likely that the mechanoreceptors in the skin of sea snakes cannot detect pressure, but can detect motion displacement, velocity or acceleration, similar to the mechanosensitive neuromasts in the lateral line of fishes and other animals that detect motion, but not pressure. We briefly mention the possibility that scale organs are sensitive to pressure (i.e. baroreception) in the Discussion (lines 436-442).
Related to the above, note that the Merkel cell neurite complexes characteristic of crocodilian ISOs are also mechanosensory and hence would not actually play an "alternative sensory role" (p. 4, lines 94-95).	Indeed, Merkel cell neurite complexes are associated with mechanoreception. We have modified text to reflect this (lines 89-95):
	"We aimed to better understand the evolution of scale organs in sea snakes by describing their ultrastructure in two fully-aquatic species, Aipysurus laevis and Hydrophis stokesii, using immunohistochemistry, and light and electron microscopy. If sea snake scale organs are retained for a mechanosensory role, either close-contact touch or detection of water motion (deflected off objects or prey/predators), we would expect them to have retained the ultrastructure described in terrestrial snakes,

	and possibly contain other sensory cells such as the Merkel-cell neurite complexes of crocodilian ISOs.”
Finally, it is suggested (p. 4, lines 80-82) that Acrochordus uses its scale organs to sense “water motion.” I do not believe this is true. My understanding is that the snakes respond to physical touch of the fish. This should be checked.	We have checked the original reference (Povel and van der Kooij, 1997) and subsequent refs (Lillywhite 2014), in which the authors confirm mechanosensory function to scale ‘sensillae’ in Acrochordus and speculate that they are sensitive to water motion generated by fish prey given its analogous structure to hair-cells within fish neuromasts. We have modified the text to reflect the uncertainty in function in the literature. In the introduction (lines 80-83):
	“Indeed, two independently aquatic snakes, Erpeton and Acrochordus , are distantly related to hydrophiine sea snakes but have protruding organs that are likely sensitive to water motion generated by the movement of fish prey (5,35).”
	In the Discussion (lines 373-374):
	“Scale ‘sensillae’ in file snakes (Acrochordus) are thought to be sensitive to the hydrodynamic motion generated by the movement of fish prey.”
(2) Throughout the paper the scale organs are referred to as “sensilla”. I understand that this term is often used in the literature to refer to similar structures in squamate reptiles. However, it is inaccurate and misleading as a sensillum is a hair-like structure (usually in insects or other invertebrate taxa). It is also true that many lizard scale organs have a central hair-like protrusion that could more accurately be called a sensillum (exclusive of the rest of the receptor). Other terms are used in the literature for the sensory organs described in this paper, including ‘scale organ’ and ‘integumentary sensory organ’ or ‘ISO’. These would seem to me to be far preferable than “sensilla/sensillum”, but obviously this is the choice of the authors.	We agree with Review 2 and welcome the opportunity to use a more accurate term. After defining the term in the introduction (lines 15-16), we have replaced ‘sensilla/sensillum’ with ‘scale organ/s’ throughout the ms.
(3) Throughout the description of the sensory organs its inner, dermal component is referred to as a “dermal capsule”. In my opinion, this is inaccurate and inconsistent with most anatomical usage. A capsule represents a discrete covering or sheath that surrounds something [“a membranous structure...that envelops an organ”— Stedman’s Medical Dictionary , 27th ed.], which is not the case here. The dermal protrusion into the epidermis is more accurately described as a ‘dermal	Concurring with Reviewers 2 point above (2), ‘dermal capsule’ is anatomically inaccurate term for the underlying organ structure that we describe in the ms. Accordingly, we have replaced all instances of ‘dermal capsule’ with ‘dermal papilla’ throughout the ms.

papilla’, as parenthetically noted on p. 5, line 134. This should be the term used throughout the paper.	
(4) Note that a finding of tactile/mechanoreception does not exclude all other possible functions, e.g., they could still function in modifying flow over the snake’s surface (though unlikely).	We have added the following sentence to paragraph ‘Dermal photoreception and other cutaneous sensory modalities’ in the discussion to address this point (lines 448-450):
	“Finally, these sensory hypotheses do not exclude other non-sensory functions for scale organs, e.g. modifying boundary layer of skin, so these roles should be considered in future studies in the scale organs of sea snakes.”
EDITORIAL COMMENTS	
Line 40: I do not believe that there is any evidence to support the assertion that snakes use mechanoreceptors to discriminate prey types. This is pure speculation based on mixed receptor types within the mouth. Evidence suggests that virtually all prey discrimination is chemosensory (gustatory and vomeronasal).	The paper referenced in the text (Nishida et al. 2000) describes ultrastructure of papillae in the mouth of Elaphe snakes, which provides compelling evidence for both chemo and mechanoreceptive functions for these oral organs. Nevertheless, we agree that the evidenced that these mechanoreceptors are used to discriminate prey types is indeed speculative. We have changed the text to clarify that these organs may be used in feeding (as appose to discriminating prey types) (lines 39-41):
	“Snakes are likely to use these mechanosensory organs to explore and navigate substrate (7,8), during courtship (11) and feeding behaviours (9,10), but the anatomy and neurophysiology of scale organs are conspicuously understudied in comparison to other sensory organs”
Line 45: change “stimulus” to ‘stimuli’	This change has been made.
Line 95: delete hyphen in “Merkel-cell”	This change has been made throughout the ms.
Line 104: “collected 1 10 km offshore”; seems to be a typo, not clear what it should be	A hyphen was deleted during the conversion to pdf, I have changes to ‘1 to 10 km offshore’
Line 127: and elsewhere; insert ‘trichrome after “Gomori’s one-step”; also, do not capitalize “One-Step”	This change has been made.
Line 130: delete “the height (thickness)” and change to ‘thickness’	This change has been made.
Line 184: It is unclear what is meant by “horizontally arranged” in reference to the ‘central cells’; in the images they are either clustered, vertical or circular—never horizontal	We have removed this term from the text.
Line 187: re: the apparently basal taper in H. stokesii —are you sure that this is not simply a plane of section issue? Did you have serial sections	I have re-examined my images of serial sections and have come to the same conclusion as the reviewer: the basal taper is likely the result of the plane of the section. We

across the width of the receptors to confirm this? From other images it appears that the dermal papilla extends outward/laterally toward the scale surface, i.e., the distal part of the papilla is wider than its base. As such, a section that just passes through the point that a lateral extension joins the central core would look tapered toward the base.	have made the following change to the ms (lines 189-190): “‘The dermal papilla was occasionally tapered at its basal end in H. stokesii (Figure 3A), but this likely to be an artefact of tissue sectioning.’”
Line 204: change “skin” to ‘scale’ for clarity	This change has been made.
Line 238: change “present at base of dermal” to ‘present at the base of the dermal’	This change has been made.
Line 247: replace the comma with a period; start new sentence with “The outer bumps...”	This change has been made.
Line 279: The comment about “cap cells” based on Jackson (1977) seems a bit pointless. There is no histological difference between the keratinocytes covering the dermal papilla and others. Obviously any cells in this position would provide abrasion resistance, but no moreso than anywhere else	We agree with this comment and have removed the term ‘cap cells’ throughout the ms and referred to them simply as ‘the keratinocytes above the dermal papilla’ as necessary. However, we have kept the term ‘cap cells’ as Jackson (1977) is one of the only available previous studies that have described scale organs in snakes and so we believe the terminology should be noted in the ms.
Lines 286-287: The figures do not show any direct evidence of discoid receptor innervation that I can see. Obviously they must be innervated and the nerves get pretty close within the scale organ, but no nerves leading directly to the discoid receptors, particularly the more distant, epidermal receptors, are evident.	Indeed. Previous studies on snake skin have found that nerves originating in the dermis terminate in epidermal ‘discoid’ receptors (Proske 1969). However, we did not observe direct evidence in our serial sections of this, therefore the text has been changed accordingly (lines 214-216): “‘Dermal axons travelled to the scale organs (Figure 5C), then meandered through the central dermal papilla before innervating the outer epidermis and presumably terminate as distinct discoid endings in the alpha layer (Figure 5A, B).’”
Lines 296-299: It seems implausible to me that the central cells have no sensory function. What’s the point of the whole organ structure then, particularly given that discoid receptors are distributed all over? How confident are you about this, i.e, what is the probability that you would have seen synaptic complexes? There are neurons all over within the dermal papilla and it seems unlikely that they are merely supplying discoid receptors. What else would they be doing? Just free nerve endings?	We concur with Reviewer 2, this sentence now reads as such (lines 301-304): “‘We did not find synaptic contacts between axons and central cells, which is consistent with light microscopy studies of other colubroid snakes (e.g. Elaphe) (23). Nevertheless, the presence of discoid receptors superior to the dermal papilla suggest that the central cells have a functional role in transducing mechanical stimuli.’”

Lines 359-362: Catania (1995), ref. 59, does not provide any evidence for sensitivity to hydrodynamic stimuli in star-nosed moles, nor can I find any other reference to such a thing. They definitely use the star organ for direct touch of food objects while foraging within water, but again (see General Comment 1), this is no different from terrestrial touch. I did not check ref. 60 for the platypus, but I would confirm that they do, indeed, have receptors that are sensitive to water movement AND that they have been “co-opted” from terrestrial cutaneous touch receptors. In fact, I would confirm this for all of them. Direct touch underwater is not the same thing as being used to detect water movement (hydrodynamic stimuli). Pinniped whiskers are a good example for mammals (they use them to detect vortices indicating fish trails) [see review by Dehnhardt & Mauck (2008), pp. 295-314, In *Sensory Evolution on the Threshold*, JGM Thewissen and S. Nummela (eds.), Univ. of California Press, Berkeley, CA]. also manatee whiskers and body hairs [reviewed in Bauer et al. (2018) *The tactile senses of marine mammals*. *Int. J. Comp. Psychology* 31, special issue, M. Botero ed.].

We have deleted these references from the text and included pinniped vibrissae as a prime example of co-option of cutaneous mechanoreceptors.

Line 401: insert ‘compared to a’ after “differential sensitivity”

This change has been made.

Line 402: I don’t see why the trigeminal or other cranial nerves that innervate cephalic cutaneous receptors are “specialized”... The sensory organs they innervate might be specialized receptors, but the cranial nerves, themselves, are not specialized.

This line has been deleted, and new sentence has been added to add clarity (lines 414-419):
 “Future studies should investigate the neural pathways and compare electrophysiological responses underlying scale mechanoreceptors distributed on the head and body of snakes. Such efforts may discover that sea snakes possess specialised nerve pathways and/or responsive fields that are analogous to the cranial nerve canals of neuromasts in fish and amphibians, or the vibrissae of secondarily-aquatic systems in mammals (55,61), which would support a hydrodynamic function for cephalic scale organs.”

Line 404: the “dorsal root ganglion” is not a “peripheral nerve of the spinal cord”—it is a part of the spinal nerve within which the sensory nerve bodies lie. For lines 402-404, it is sufficient to note that the cephalic receptors are innervated by cranial nerves while the

This line has been deleted (see change in text in previous point).

postcranial receptors are innervated by spinal nerves, which is exactly what one would expect.	
Line 406: the receptors are not used “to actively seek”—they are used while he snake actively seeks...	This line has been deleted (see change in text in previous point).
Line 409: do you mean ‘electrophysiological’ rather than ‘electrophysical’?	Indeed, this change has been made.
Lines 429-432: there is also no histological support for a magnetic sense	We have changes this line to make a stronger assertion in line with Reviewer 1 and Reviewer 2 comments on magnetic sense (lines 432-436):
	“Several other sensory functions have been tentatively attributed to the scale organs of sea snakes, but these currently lack supporting evidence. An electro-magneto-sense is plausible (5), but our histological sections do not show canals or pores that are indicative of passive electroreceptors (e.g. ampullary-type organs) or specialised active electroreceptive organs (e.g. tuberous organs or mormyromasts of weakly-electric fish) (66,67).”
FIGURE LEGENDS: 2A: no plane of section is give for the image; 2B: delete “cross-“ (redundant); ‘hematoxylin’ is misspelled “hemotoxylin”; 2C: insert ‘trichrome’ after “Gomori one-step”; 3A: no plane of section given; 3B,C: delete “cross-“; ‘transverse’ is misspelled “traverse”; 4: check for above changes; also, Latin binomial name not italicized; “deeper cross-section” doesn’t make sense to me, please clarify; 7: “keratin filaments tonofilaments (t)” must be a typo—missing parentheses?	These changes have been made.

Appendix B

**Ultrastructural evidence of a mechanosensory function of scale organs ('sensilla') in sea**
**snakes (Hydrophiinae)**

Jenna M. Crowe-Riddell^{1*}, Ruth Williams², Lucille Chapis³, Kate L. Sanders¹

¹School of Biological Sciences, The University of Adelaide, Adelaide SA 5005, Australia

²Adelaide Microscopy, the Centre for Advanced Microscopy and Microanalysis, Adelaide SA 5005, Australia

³College of Life and Environmental Science, University of Exeter, Exeter EX4 4QD, United Kingdom

*Corresponding author: jenna.crowe-riddell@adelaide.edu.au ajmcroweriddell@gmail.com

Abstract

The evolution of epidermal scales was a major innovation in lepidosaurs, providing a barrier
to dehydration and physical stress, while functioning as a sensitive interface for detecting
mechanical stimuli in the environment. In snakes, mechanoreception involves tiny scale
organs ('sensilla') that are concentrated on the surface of the head. The fully marine sea
snakes (Hydrophiinae) are closely related to terrestrial hydrophiine snakes but have
substantially more protruding (dome-shaped) ~~sensilla~~scale organs that often cover a larger
portion of the scale surface. Various divergent selection pressures in the marine environment
could account for this morphological variation, ~~including relating to enhanced~~ detection of
mechanical stimuli ~~from direct contact with stimuli and/or indirect contact via water motion~~
~~(i.e. 'hydrodynamic reception' either tactile or hydrodynamic)~~, or co-option for alternate
sensory or non-sensory functions. We addressed these hypotheses using
immunohistochemistry and light- and electron microscopy to describe the cells and nerve
connections underlying scale ~~sensilla~~organs in two sea snakes, *Aipysurus laevis* and
*Hydrophis stokesii*. Our results show ultrastructural features in the cephalic ~~sensilla~~scale
organs of both marine species that closely resemble the mechanosensitive Meissner-like
corpuscles that underlie terrestrial snake ~~sensilla~~scale organs. We conclude that the
~~sensilla~~scale organs of marine hydrophiines have retained a mechanosensory function, but
future studies are needed to examine whether they are sensitive to hydrodynamic stimuli.

**Keywords:** sea snake, ~~sensilla~~scale organs, cutaneous, mechanoreceptor, skin, ultrastructure,
transmission electron microscopy, sensilla

Introduction

Hardened epidermal scales are a characteristic trait of snakes (and other lepidosaurs: lizards
and tuatara) that facilitate defensive signalling, camouflage, water retention, and locomotion
(1–3). The epidermal scales also provide the primary surface for mechanoreception, which is
the ability to sense mechanical stimuli that result from ~~pressure or~~ physical displacement
(vibration) (4). Scale organs ('sensilla' ~~or~~ 'tubercles' *sensu* (5–7) are small
mechanoreceptors that protrude from the surface of epidermal scales of the head and body of
snakes. Snakes are likely to use these mechanosensory organs ~~to~~ explore and navigate
substrate (8,9), ~~discriminate prey types (9,10) and during engage in courtship behaviours~~

(10), and feeding (11,12) behaviorus, but the anatomy and neurophysiology of scale
sensillaorgans are conspicuously understudied in comparison to other sensory organs (~~e.g.~~,
for example, eyes (13), ~~+~~auditory structures (14), vomeronasal organ (15) and, heat-sensing
pits (16).

In terrestrial snakes, scale sensillaorgans are concentrated on the head and are highly
sensitive to mechanical stimulation, particularly moving stimuli~~s~~ (17–20). The underlying
ultrastructure of cephalic scale sensillaorgans consists of an innervated cluster of dermal cells
(~~‘dermal papillaeapsule’~~) that displaces the surrounding epidermis to create round skin
elevations (21–23). These underlying features of scale sensillaorgans have been likened to
‘Meissner corpuscles’, which are low-threshold mechanoreceptors (LTMRs) sensitive to
innocuous (~~‘light touch’~~) stimuli in the glabrous (hairless) skin of mammals (24,25). Scale
sensillaorgans on the ~~body-head~~ are ~~less-more~~ specialised in their underlying ultrastructure
than on the body, which- they lack dermal ~~capsulepapillae-~~ and the outer skin elevations are
instead caused by a superficial thickening of the epidermis (10,26). ~~These ultrastructural~~
~~differences between head and body scale sensilla, and the concentration of sensilla on the~~
~~head, are thought to reflect the role of the head as the primary tactile interface in snakes~~
~~(17,27).~~

Snakes exhibit substantial variation in the size, shape, density, and distributions of their
scale sensillaorgans. Enlarged and/or high densities of sensilla scale organs have been
reported in fossorial snakes (*e.g.* Leptotyphlopidae) and some sea snakes (Hydrophiinae),
whereas in other colubroid snakes sensillaorgans are small and/or sparse (*e.g.* Dipsadinae) or
even absent in some species (*e.g.* Viperidae) (22,27–29). Interspecific differences in the traits
of these sensillascale organs ~~traits~~ likely relate to various aspects of species’ environment,
ecology, and phylogeny. However, our understanding of the adaptive diversity ~~functional~~-of
snake sensillascale organs is hindered by a lack of comparative data describing differences in
the external traits of scale sensillaorgans-traits and their underlying ultrastructure.

Hydrophiine snakes (Elapidae) provide a useful comparative framework to investigate
the evolution of squamate scale sensillaorgans in response to major ecological transitions (7).
The ~~fully marine~~, viviparous sea snakes comprise a clade of more than 60 species that
evolved within the terrestrial Australian hydrophiine radiation (tiger snakes, death adders,
taipans) approximately 9 to 18 million years ago (30). Previous work has found that the
cephalic scale sensillaorgans of sea snakes are substantially more protruding (dome-shaped)
compared to their terrestrial counterparts, and in some lineages cover a much larger
proportion of the scale surface (> 6% versus < 2.5% in sampled taxa) (7).-. This divergence in

Formatted: Font: Italic

[revised manuscript text omitted]

**Cephalic scale organs**

Observed under a stereomicroscope, the cephalic scale sensillaorgans appeared as
unpigmented external elevations ('bumps') of outer skin (Figure 1). Observed under light
microscopy, the cephalic scale sensillaorgans of *A. laevis* (Figure 2) and *H. stokesii* (Figure
3) shared a similar structure that consisted of a cluster of 9 to 11 cells ('central cells'), ~~which~~
~~were horizontally arranged~~, originating in the dermis and ~~evaginated-evaginating~~ the
epidermis to create a dermal eapsulepapilla ('papilla'). The ratio of length to diameter of the
dermal eapsulepapilla was approximately 1:1 for both *A. laevis* and *H. stokesii* (Table S1).
The dermal eapsulepapilla was occasionally tapered at its basal end in *H. stokesii* (Figure
3A), ~~but-but this likely to be an artefact of tissue sectioning. remained expanded in *A. laevis*~~
~~(Figure 2B, C)~~. In some dermal eapsulespapillae we were able to identify a blood vessel
leading to (and thus presumably vascularising) the central cells (Figure 2B). In *H. stokesii*,

Commented [JC2]: Not technically 'keratinised' as hardened structure is not derived from 'keratins' – updated ref

the Gomori's One Step Gomori's one step trichrome stain revealed collagen fibres
interspersed between central cells and often separated the dermal eapsulepapilla from
keratinocytes within the epidermis (Figure 3C). In both species, the dermal eapsulepapilla
displaced surrounding epidermal layers so that the columnar cells of the *stratum*
*germinativum* were positioned above the dermal eapsulepapilla, causing the bumps of the
outer skin surface (Figure 2; Figure 3). In *A. laevis*, the epidermis above the dermal
eapsulepapilla ('cap cells') was approximately 50% thinner than the epidermis of the
surrounding regions of skin that did not contain sensillaorgans (17 μm ; $t = -11.16$, 110 d.f., P
< 0.001) and in *H. stokesii*, it was approximately 15% thinner than the adjacent flat epidermis
(28 μm ; $t = -2.19$, 67 d.f., $P = 0.03$).

There was a second type of dermal eapsulepapilla on the cephalic scales in *H. stokesii*
that contained approximately 10 central cells and displaced the surrounding epidermis but, in
contrast to the cephalic scale sensillaorgans, did not result in a distinctive bumps in the outer
skin surface (Figure 3.4). These smaller scale sensillaorgans were more variable in shape
compared to typical sensillaorgans (ratio length:diameter 1.7; Table S1) and often located at
the base of depressions on the outer surface of the scalekin (Figure 4). The epidermis above
the dermal eapsulepapilla was 25% thinner than adjacent flat epidermis (25 μm , $t = -2.76$, 26
220 d.f., $P = 0.01$; approximately same height as cap cells the epidermis above papilla of other
cephalic sensillaorgans, $t = 0.85$, 12 d.f., $P = 0.41$).

The cephalic dermis and epidermis of *H. stokesii* were immunoreactive for PGP9.5
(Figure 3.5). Specificity of immunoreactions were confirmed by antibody controls (Figure
S1) and by the localised staining of nerve bundles that had previously been identified under
light microscopy (Figure 2A; Figure 3A). Dermal axons travelled to the scale sensillaorgans
(Figure 5C), then meandered through the central dermal eapsulepapilla before innervating the
outer epidermis and often (presumably) terminating as distinct discoid endings in the alpha
layer (Figure 5A, B). These discoid endings were primarily located above the dermal
eapsulepapilla, but were also present in flat epidermis that did not contain sensillaorgans
(Figure 6A). Unfortunately, the second type of dermal eapsulespapillae in *H. stokesii*
(described above; Figure 4) were not present in the sections stained for
immunohistochemistry.

Immunoreactions were also localised to ovoid structures within the cephalic dermis of
*H. stokesii* (Figure 3.6). These structures corresponded to lamellar cells that were ovoid in
shape and resembled small Pacinian-like corpuscles (mean length $29 \pm 15 \mu\text{m}$ and mean
diameter of $22 \pm 12 \mu\text{m}$; Table S1). The location of these 'lamellar corpuscles' in *H. stokesii*

ranged from 61 to 124 μm (mean 93 μm) depth from the basal layer of the epidermis.
Lamellar corpuscles were also identified in the cephalic dermis of *A. laevis* that were a
similar ovoid shape (mean length $37 \pm 26 \mu\text{m}$ and mean diameter $25 \pm 5 \mu\text{m}$; Figure 2C) to
those found on the cephalic dermis of *H. stokesii*. The location of the lamellar corpuscles in
*A. laevis* ranged from 53 to 168 μm (mean 118 μm ; Table S1) depth from the basal layer of
the epidermis. Although the lamellar corpuscles were dispersed throughout the dermis
(*stratum laxum*), they were often subjacent to scale *sensillaorgans* (Figure 2C; Figure 3B,C).
Unfortunately, due to preservation issues we were unable to perform immunohistochemistry
on these cephalic skin sections in *A. laevis*.

The dermal *capsulepapilla* of a scale *sensillumorgan* was observed in *A. laevis* using
electron microscopy (Figure 7). High magnification images showed a cluster of central cells
within the dermal *capsulepapilla* (Figure 7B). These central cells were distinguished from
surrounding keratinocytes by their round shape and lack of tonofilaments (Figure 7B inset
two). Tonofilaments were present in the intracellular space of keratinocytes throughout the
epidermis (Figure 7B). Tight junctions (desmosomes) and associated tonofibrils can be seen
between central cells and keratinocytes (Figure 7B inset two. In the intercellular domain,
small bundles of transverse collagen fibres and a single, small putative nerve axon were
present at base of the dermal *capsulepapilla* (closer to dermis; Figure 7B inset one). Small
phospholipid inclusions were also present (Figure 7B inset one). Unfortunately, we were
unable to image the putative axon at higher magnification so could not confirm the presence
of neuronal elements (*e.g.* lamellar arrangement of Schwann cells, neurofilaments).

**Scale organs on the tail of *Aipysurus laevis***

Two scale structures were identified in the tail skin of *A. laevis*. Although, we were unable to
discern bumps in the outer tail skin surface using a stereomicroscope (Figure 1B), several
skin elevations were identified in cross-sections of the skin under light microscopy (Figure
8). The epidermal elevations of the tail ('tail scale *sensillaorgans*') lacked the dermal
*capsulespapillae* associated with the cephalic scale *sensillaorgans*. The outer bumps were
instead created by thickening of the epidermis (Figure 8A), which was 57% thicker than
adjacent flat epidermis (47 μm , $t = 14.18$, 86 d.f., $P < 0.001$) and 17 μm (65%) thicker than
~~cap cells~~the epidermis above of cephalic scale *sensillaorgans* ($t = -14.26$, 18 d.f., $P < 0.001$).
Tail scale *sensillaorgans* also lacked the collagen fibres and blood vessels that were
associated with cephalic scale *sensillaorgans*. A second scale structure identified in the tail
skin of *A. laevis* consisted of a small dermal *capsulepapilla* of approximately 10 central cells

with a ratio of length and diameter of 1:1 (Table S1; Figure 8B). Although the dermal
capsulepapilla displaced the surrounding epidermal layer (including the columnar cells of the
*stratum germinativum*) this did not result in elevations of the outer epidermis (Figure 6B).
The epidermis above the dermal capsulepapilla was 42% thinner than adjacent epidermis that
did not contain dermal capsulespapillae (11 μm ; $t = -4.65$, 86 d.f., $P < 0.001$) and slightly
thinner (5.5 μm) than the ~~cap cells~~epidermis above of cephalic scale sensillaorgans ($t = 2.59$,
18 d.f., $P = 0.02$). Subjacent to these tail dermal capsulespapillae, collagen fibres in the
dermis (*stratum laxum*) were dispersed and melanosomes could not be seen (Figure 8B).
Unfortunately, due to preservation issues we were unable to perform immunohistochemistry
on these tail sections.

Discussion

Cephalic scale organs

*Scale sensillaorgans and dermal capsulepapillae*

Previous work found that the cephalic sensillaascale organs of sea snakes are substantially
more protruding and often cover a larger proportion of the scale surface than the sensillaascale
organs of terrestrial hydrophiine snakes (7). The present study shows that, despite these
differences, sensillaorgans in sea snakes have retained a similar underlying ultrastructure to
their terrestrial counterparts. The sensillaascale organs examined in *Aipysurus laevis* and
*Hydrophis stokesii* are characterised by a dermal capsulepapilla that consists of an
aggregation of central cells with collagen fibres, blood vessels and nerve axons in the
intercellular domain that together displace the surrounding epidermis (Figure 2-7). A similar
underlying structure has been reported for the cephalic scale sensillaorgans of ~~ten~~terrestrial
species representing ~~the several major phylogenetic~~ groups of ~~snakes~~ squamates ~~including~~
~~(agamids, henophidians, iguanids, scolecophidians, and colubroids,)~~ ~~and lizards~~ (agamids,
iguanids and varanids) (10,21–23,42–45). In terrestrial snakes and sea snakes, the epidermis
above the dermal capsulepapilla is comprised of columnar keratinocytes (i.e. *stratum*
*germinatum*) that form a layer that is 15% to 50% thinner than the epidermis of adjacent
flat skin. The columnar keratinocytes above the dermal capsulepapilla have been described as
‘cap cells’ in snakes and suggested to provide protection against abrasion or aid in
transducing mechanosensory stimuli (22).

We discovered that sea snake skin contained ~~free~~nerve axons that extend from the dermis
and terminate within the alpha layer (epidermis) as distinct discoid structures (Figure 35). In

Commented [JC3]: ‘Squamates’ as snakes and lizards are not monophyletic groups

terrestrial colubroid snakes, these structures have variously been described as ‘discoïd
receptors’ (21), ‘end bulbs’ (20) and ‘button-like’ (10) nerve endings. In the sea snake skin,
we found discoïd receptors distributed throughout the epidermis, but aggregated above the
dermal capsulespapillae (Figure 5A, B) deriving from axons at the base of the dermal
capsulepapilla (Figure 3.5C). This adds evidence for a sensory function of scale
sensillaorgans in sea snakes.

Our images from transmission electron microscopy provide the first high resolution
ultrastructure data of a cephalic scale sensillum-organ in a snake. Inspection of Figure 3.7B
shows that the central cells within dermal capsulepapilla are clearly differentiated from
surrounding keratinocytes by their lack of tonofilaments. Tonofilaments are keratin
formations of keratin-like proteins that provide structural integrity to the squamate epidermis
(46). Although lacking in tonofilaments, central cells maintain contact elements with
surrounding keratinocytes via multiple tight junctions (desmosomes) (Figure 7B, inset two).
A putative axon was also identified in the intercellular domain of the dermal capsulepapilla
(Figure 7B, inset one), which may represent the ‘terminal receptors’ or myelinated axons
previously identified in lizards (45). We did not find synaptic contacts between axons and
central cells, which is consistent with light microscopy studies of other colubroid snakes (*e.g.*
*Elaphe*), (23). Nevertheless, the presence of discoïd receptors superior to the dermal papilla
suggest that the -and suggests that the central cells - (and associated dermal capsule)- have a
functional role in transducing mechanical stimuli. have a structural role rather than
functioning as a direct transducer of stimuli.

In addition to dermal capsulespapillae associated with cephalic scale sensillaorgans, we
detected capsulespapillae typically (but not always) located at the bottom-base of depressions
in the outer skin in *H. stokesii* (Figure 4). These dermal capsulespapillae consisted of
approximately 10 central cells (Figure 5) and displaced surrounding keratinocytes but, in
contrast to the ultrastructure we describe for cephalic scale sensillaorgans, did not result in a
skin elevation (bump). Putative nerve structures leading to the dermal capsulepapilla were
identified using Gomori’s One Step Gomori’s one step trichrome stain under light microscopy
(Figure 4B), but we were unable to conduct antibody staining of neuronal markers. It is
unclear whether these dermal capsulespapillae are distinct scale structures or merely
undeveloped or damaged scale sensillaorgans.

*Lamellar corpuscles*

Commented [JC4]: Correct this

We detected lamellated, ovoid cells in the deeper dermis of cephalic skin in both species
examined and demonstrated that these lamellar corpuscles were neuronal-positive in *H.*
*stokesii* (Figure 6). These structures resemble the ‘non-encapsulated lamellated receptors’
identified in other squamates such as ~~(*Boa*, *Elaphe*, *Iguana*, and *Agama* genera)~~
(11,12,21,47). The location and shape of these receptors suggest that they are small, Pacinian-
like corpuscles (24,48). Pacinian (Vater-Pacini) corpuscles are rapidly adapting LTMRs that
are sensitive to skin indentation and vibratory (‘deep touch’) stimuli of high frequencies
(peak 250 Hz, range 40 to 800 Hz), and they are present in glabrous skin of mammals (24).
Pacinian corpuscles consist of connective tissue and fibroblasts lined by flat neuronal
‘Schwann’ cells; the lamellar structures identified in sea snakes tested immuno-positive for
the neuronal maker PgP9.5 suggesting that these are indeed modified neuronal cells. The
sensitivity of these receptors has not been ~~targeted-examined~~ in previous electrophysiological
tests ~~of~~ snakes skin.

*Ancestral and derived sensory functions for cephalic scale organs*

The ultrastructural features described above for cephalic scale sensillaorgans of terrestrial and
marine snakes represent all of the components of Meissner-like corpuscles. Meissner
corpuscles are rapidly adapting LTMRs present in the dermal papillae of mammal glabrous
skin (48). Electrophysiological experiments of cranial nerves in colubroid snakes found that
they are rapidly adapting LTMRs with receptive fields that overlap with Meissner corpuscles
(*i.e.* 12 mm², (17). Our finding that the cephalic sensillascale organs of sea snakes share a
very similar ultrastructure with their terrestrial relatives (and appear to lack novel or
specialised cell types) provides evidence that marine lineages have retained the ancestral
mechanosensory role for these organs.

The dome-shape and often high scale-coverage of ~~scale~~ mechanoreceptors in sea snakes
suggests divergent selection on these organs in marine environments, either for retained
(ancestral) enhanced sensitivity to tactile stimuli or a derived sensitivity to hydrodynamic
stimuli. Sea snakes forage in benthic habitats, frequently probing burrows and crevices as do
terrestrial snakes on land (49), ~~but there~~ ~~There~~ is no obvious reason ~~that~~ why sea snakes
should require a heightened tactile sense compared to terrestrial species. ~~Sea snakes forage in~~
~~benthic habitats, frequently probing burrows and crevices as do terrestrial snakes on land~~
~~(51)~~. It seems more likely that sea snakes have experienced selection pressures for sensitivity
to hydrodynamic stimuli (7). Observations of the sea snake *Hydrophis (Pelamis) platurus*
approaching and biting a vibrating object (50) provides some behavioural evidence that sea

Formatted: Indent: First line: 0 pi

snakes are responsive to hydrodynamic stimuli. Evoked potentials have been recorded from
the midbrain of the sea snake *Hydrophis (Lapemis) curtus* in response to a vibrating sphere
(50 to 200 Hz, peak sensitivity at 100 Hz), but no nervous response was successfully
recorded directly from a scale ~~sensillumorgan~~ ~~of this species~~ (51). However, more recently,
auditory evoked potentials were recorded ~~(from the midbrain)~~ of *A. laevis* and *H. stokesii* in
response to tone bursts from 40 to 600 Hz (peak sensitivity at 60 Hz) (Chapuis *et al.*,
~~unpublished data~~ ~~under review~~). ~~These preliminary investigations~~ This work showed that some
species of sea snakes are capable of detecting low amplitude water motion, ~~and~~ pressure
and/or particle motion ~~caused by sound stimuli~~. ~~Moreover, although~~ ~~T~~ these studies were not
able to discern whether ~~hydrodynamic stimuli~~ water motion ~~were~~ as being received detected
by mechanoreceptive scale organs in the skin or hair-cells in the inner ear. ~~However~~, the
peak sensitivities to ~~the~~ mechanical stimuli broadly overlap with peak sensitivities of
Meissner (10 to 50 Hz) and Pacinian (200 to 300 Hz) corpuscles.

Hydrodynamic reception allows the detection of water motion, usually caused by water
disturbances or animal movement, and is characterised by very low frequency components
~~that~~ (peak at 10 Hz ~~with a~~ ~~maximum of~~ 50 Hz. ~~This sensory ability~~ ~~It~~ has evolved
repeatedly in aquatic organisms ~~(e.g. the lateral line systems in fish, cephalopods and~~
~~amphibians)~~ wherein hydrodynamic stimuli are transduced by cutaneous mechanoreceptors
~~(e.g. the lateral line systems in fish, cephalopods and amphibians~~ (52,53). Cutaneous
mechanoreceptors have also been co-opted for hydrodynamic reception in ~~aquatically~~
~~foraging animals~~ ~~secondarily-aquatic-including~~ mammals, a well-studied example of which
~~are the vibrissae (whiskers) of pinnipeds~~ (54–56), ~~(e.g. star nosed moles~~ (58), platypus (59),
~~birds (e.g. ducks, geese, ibis~~ (60) and reptiles (61).

Among snakes, two independently aquatic taxa (that are distantly related to hydrophiines)
have evolved highly derived scale mechanoreceptors ~~that putatively function to~~ sense the
water motions generated by the movement of prey. Tentacled snakes (*Erpeton tentaculum*)
have the largest mechanoreceptors among vertebrates with two cephalic tentacles (measuring
two to three 3 millimetres) and made up of dermis, epidermis and free nerve endings
(34,57). Scale 'sensillae' in file snakes (*Acrochordus*) are thought to be sensitive to the
hydrodynamic motion generated by the movement of fish prey. These small organs are
vascularised like sea snake ~~scale sensilla~~ organs but instead of a dermal ~~capsule~~ papilla they
consist of specialised epidermal cells that underlie highly-derived bristles that protrude from
the skin (5,15). ~~Tentacled snakes (*Erpeton tentaculum*) have the largest mechanoreceptors~~
~~among vertebrates with two cephalic tentacles (2 to 3 millimetres) made up of dermis,~~

~~epidermis and free nerve endings (35,59).~~ *Erpeton* and *Acrochordus*. These snake lineages
represent older aquatic transitions in snake phylogeny, and their mechanoreception is linked
to specialised ~~ambush predator~~ strategies for ~~hunting-ambushing fish prey~~ in turbid
freshwater habitats of low visibility (58). In contrast, sea snakes have recent marine origins
and are an ecologically very diverse clade comprising species that variably occupy blue
water reefs or turbid inshore habitats, are diurnal or nocturnal, and specialise on active or
sedentary prey. The turtle-headed sea snake, *Emydocephalus annulatus*, is notable in having
the second highest scale coverage of sensilla scale organs (3.8%) while being diurnally active
and specialising on sessile fish eggs in clear water reefs (7,59). ~~These results suggest show as~~
a complex evolutionary history of scale organs and further research should aim to link
selection pressures on hydrodynamic reception with particular ecologies (e.g. that optimal
foraging strategies, water turbidity) may not be the primary selection pressure for
hydrodynamic sense in among sea snakes.

Formatted: Font: Italic

Formatted: Font: Italic

Formatted: Font: Italic

Tail scale organs

Based on cellular morphology, there is a clear distinction between cephalic and posteriorly
located scale sensilla organs in sea snakes. Scale sensilla organs present on the tail skin of *A.*
*laevis* do not contain dermal capsules papillae; skin elevations are instead created by a
thickening of the epidermis (Figure 8A). These structurally ‘simplified’ sensilla
structures scale organs have been reported in ~~studies of~~ the body skin of the sea snake *E.*
*annulatus* (37) and the tail skin of some terrestrial snakes (10). Many functional roles have
been proposed for body scale sensilla organs in sea snakes, including mechanoreception, sex
mate recognition, and enhanced friction for improved swimming performance, gripping
and/or ecdysis (7,26,37). We were unable to stain for the presence of free nerve endings in
tail scale sensilla organs, however, nerve staining of ‘supraclacal tubercles’ in the snakes
*Thamnophis sirtalis* and *Nerodia rhombifer* (formally *Natrix rhombifera*) found that they
were innervated in a similar pattern to cephalic scale sensilla organs, and thought to be
important for sensory feedback to aid alignment of the cloacae position during copulation
(10,60). ~~Thus, Although these~~ posteriorly located scale sensilla organs are exhibit clearly
ultrastructural differences stiated compared to ~~from~~ cephalic scale sensilla organs, it is by their
ultrastructure and likely that they have a mechanoreceptive and/or structural functions in sea
snakes.

Formatted: Indent: First line: 0 pi

The ultrastructural differences in cephalic scale ~~sensilla~~organs *versus* posteriorly located
scale organs may reflect ~~variances~~variation in mechanoreceptor-sensitivity in the head
compared to ~~those on~~ the rest of the body. Research in mammals suggests that the structure of
the skin organ may be just as important as the neurons that carry the electrical impulse—
collagen can provide physical tethering, structural integrity, or aid in propagating or
modulating the sensation of force (25). Thus, the absence of a dermal ~~capsule~~papilla for scale
~~sensilla~~organs on the body and tail skin might indicate a less specialised mechanoreceptor
with differential sensitivity compared to a cephalic mechanoreceptor. Thus, our results
support previous studies on terrestrial snakes that suggest that the head of sea snakes is the
prime exploratory organ for actively seeking mechanical stimulation (17,27). Future studies
should investigate the neural pathways and compare electrophysiological
responses underlying scale mechanoreceptors distributed on the head and body of
snakes. Such efforts may discover that sea snakes possess specialised nerve pathways and/or
responsive fields that are analogous to the cranial nerve canals of neuromasts in fish and
amphibians, or the vibrissae of secondarily-aquatic systems in mammals (55,61), which
would support a hydrodynamic function for cephalic scale organs.

Formatted: Font: Italic

Furthermore, cephalic cutaneous receptors are innervated by specialised cranial nerves
(e.g. trigeminal ganglion), while the rest of the body is innervated by peripheral nerves of the
spinal cord (i.e. dorsal root ganglion) (4). These neural pathways are thought to reflect
differences in somatosensory processing wherein the head harbours specialised tactile
receptors that are used to actively seek stimuli in the surrounding environment, in contrast to
the body, which passively receives information (4,63). Thus, our results suggest that the head
of sea snakes is the prime exploratory organ for actively seeking mechanical stimulation.
~~Future studies should investigate the neural pathways and compare electrophysiological responses~~
~~underlying scale mechanoreceptors distributed on the head and body of snakes.~~

**Dermal photoreception and other cutaneous sensory modalities**

The skin provides a primary interface for receiving multiple stimuli, creating an opportunity
for multi-modal cutaneous receptors. Indeed, molecular and electrophysiological studies of
ISOs in crocodiles indicate multi-modal sensitivity to mechanical stimuli and thermal and pH
gradients (62,63). Dermal photoreceptors in the tail skin of *Aipysurus* sea snakes mediate
phototactic behaviour in these species (36); we did not detect candidate photoreceptive
structures (e.g. photoreceptors, stacked membranes) in the tail skin of *A. laevis*, but we did

Formatted: Font: Not Italic

find structurally simplified scale organs (described above) and other small dermal papilla
(Figure 8B). Given that cutaneous receptors have been linked with both mechano- and
photoreception in amphibians (64) and marine invertebrates (65), these scale organs merit
further investigation for their putative role in photoreception. Our study (and previous studies
using electron microscopy (Chapter 2, Crowe Riddell *et al.*, 2016; Povel and VanDerKooij,
1997) demonstrate that snake scale sensilla are devoid of pores and so a chemosensory
function is highly unlikely.

Several other sensory functions have been tentatively attributed to the scale organs of
sea snakes, but these currently lack supporting evidence. An electro-magneto-sense is
plausible (5), but our histological sections do not show canals or pores that are indicative of
passive electroreceptors (e.g. ampullary-type organs) or specialised active electroreceptive
organs (e.g. tuberous organs or mormyromasts of weakly-electric fish) (66,67). Similarly, in
addition to previous studies using electron microscopy (5,7), our study demonstrates that
scale organs in sea snakes are devoid of pores and so a chemosensory function is highly
unlikely. Baroreception of the changes in air pressure that precede extreme weather events
has been attributed to sea kraits, which are an independently marine clade of hydrophiines;
however, it is unclear whether sea snakes react in a similar way and how cutaneous
mechanoreceptors might transduce pressure information in sea snakes or sea kraits. Salinity is
an important predictor of sea snake distribution (68) because many-most species require
access to freshwater for hydration (69–71), but pH receptors are more likely to be located in
papillae in the mouth (11,72). The thermal sensitivity of scale sensillaorgans has been
investigated in *Elaphe* colubroid (*Elaphe*) snakes with results indicating, which found that
although some cutaneous nerves are exclusively sensitive to heat, mechanoreceptive fibres
are not responsive to either heating or cooling (17,73). Dermal photoreceptors in the tail skin
of *Aipysurus* sea snakes mediate phototactic behaviour in these species. Although we did not
detect candidate photoreceptive structures (e.g. photoreceptors, lenses) in the tail skin of *A.*
*taevis*, we did find ‘simplified’ scale sensilla (described above) and other small dermal
capsule (Figure 8B). Given that cutaneous receptors have been linked with both mechano-
and photoreception in amphibians (69) and marine invertebrates (70), these scale organs
merit further investigation for their putative role in photoreception. Finally, these sensory
hypotheses do not exclude other non-sensory functions for scale organs, e.g. modifying
boundary layer of skin, so these roles should be considered in future studies in the scale
organs of sea snakes.

Finally, an electro-magneto-sense has been proposed for scale sensilla in snakes (5);

Formatted: Indent: First line: 3 pi

Formatted: Font: Italic

Formatted: Font: Italic

Formatted: Font: Not Italic

Formatted: Font: Italic

~~but our understanding of the spatial ecology, and thus long range navigation abilities, of sea~~
~~snakes is limited.~~

**Conclusions**

Our study shows that the ultrastructure of cephalic ~~sensilla~~ scale organs of ~~in~~ sea snakes
closely resembles the mechanosensitive Meissner-like corpuscles that underlie the scale
~~sensilla~~ organs ~~in~~ of terrestrial snakes. This provides evidence that the sensilla scale organs of
marine hydrophiines lineages have retained an ancestral mechanosensory function. Our
findings provide the basis for future research into the sensitivity of cutaneous receptors in sea
snakes including mechano-, hydro- and photo-sensory modalities. Our study highlights that
snakes are an important group for understanding the evolution of mechanoreception in
vertebrates, particularly in response to shifting sensory landscapes.

**Ethics statement:** Animals were collected in accordance with a the Western Australian
Department of Biodiversity, Conservation and Attractions Department of Parks and Wildlife
of Western Australia licence to take fauna for scientific purposes (Permit # (SF010002).

Euthanasia was carried out in accordance with the guidelines of the Australian Code of
Practice for the Care and Use of Animals for Scientific Purposes [73], under Animal Ethics
Committee protocols from the University of Adelaide (S-2015-119) and the University of
Western Australia (RA/3/100/1369).

**Data accessibility:** Supplementary data are interactive digital scans of slides (ndp.view files)
and skin measurements (.xlsx) for *A. laevis* and *H. stokesii*. These are available with the
Supplementary Figure S1s attached separately. Available at Figshare, DOI:
10.25909/5c5bb6777f249 ; <https://figshare.com/s/f029519e4f6304b34565>

**Competing interests:** The authors declare no competing interests.

**Author contributions:** J.M.C.-R. and K.L.S. conceived of the study. J.M.C.-R., L.C. and
528 K.L.S. collected samples; ~~J.M.C.-R. and R.W. prepared samples for electron~~ carried out
microscopy analysis and interpretation with input from L.C.; J.M.C.-R. and K.L.S. wrote the-
The manuscript was written by J.M.C.-R. with significant input from all co-authors L.C. and
R.W.

**Acknowledgements:** We are grateful to Kylie Sherwood (Chelonia Broome), Caroline Kerr
(The University of Western Australia), Mick and Kelly Woodley and crew (Absolute Ocean

Charters, Broome) for assistance in ~~catching and transporting~~collecting sea snakes. We thank
Luke Allen (Venom Supplies Pty Ltd, South Australia) for supplying taipan tissue. For access
to specimens and laboratories at the South Australian Museum, we thank Mark Hutchinson
and Carolyn Kovach. We thank Kathryn Batra, Chris Leigh and Jim Manavis (Adelaide
Medical School, ~~South Australia~~University of Adelaide), Peter Hill and Lucy Woolford
(School of Animal and Veterinary Sciences, University of Adelaide), and Jane Sibbons
(Adelaide Microscopy, South Australia) for ~~assisting~~assistance with immunohistochemistry
and microscopy analyses and interpretation. We are grateful to the anonymous reviewers for
their comments that improved this manuscript.

**Funding statement:** This work was supported by a Hermon-Slade Foundation Grant
(0001039517) and Future Fellowship to K.L.S. (FT130101965), and an Australian
Government Research Training Program Scholarship and Fulbright Postgraduate Scholarship
held by J.M.C-R.

**Supplementary Material**

Figures

**Figure S1** Primary and secondary antibody controls for PGP9.5 in taipan (*Oxyuranus*
*scutellatus*) brain tissue and sea snake (*Hydrophis stokesii*) cephalic skin tissue.

Electronic files

**File ES1** Interactive digital scans of histology slides (ndp.view, zipped)

**File ES2** Skin measurements for *A. laevis* and *H. stokesii* (.xlxs).

**Table 3.1.** Taxonomy, life stage, museum accession or field numbers and sample size of two species of sea
snakes (Hydrophiinae) used in this study. Time until last shed was deduced for captive specimens by the
presence of shed skins. Tissue samples were collected from captive specimens for various microscopy analyses:
stereomicroscopy (SM), light microscopy (LM), transmission electron microscopy (TEM) and
immunohistochemistry (IHC). Museum specimens were sourced from the Western Australian Museum (WAM)
and the Field Museum of National History, Chicago (FMNH).

Taxonomy		Specimen information				Time to last shed (days)	Tissue samples	Microscopy analyses
Genus	Species	Museum or Field numbers	Sex	Life stage		Scale type & location		
Aipysurus	laevis	KLS0690	M	Adult	18	6 th supralabial (right side) Posterior tip of tail (right side)	LM LM	
	laevis	#AL270916	M	Adult	> 128	Nasal scale (right side)	TEM	
	laevis	WAMR174260	M	Subadult	Unknown	Gross morphology of skin	SM	
Hydrophis	stokesii	#HS270916A	M	Adult	107-days	Nasal scale (right side) Nasal scale (left side)	LM & IHC TEM	
	stokesii	FMNH202826	Unknown	Juvenile	Unknown	Gross morphology of skin	SM	

**Figure 1.** Gross morphology of the skin of sea snakes illustrating small, unpigmented scale organs ('sensilla').
Line drawing of sea snake indicates ~~sampling regions of available skin sampled for this study~~: nasal scales
from the head of *Aipysurus laevis* and *Hydrophis stokesii*, and supralabial scales from the head and caudal
scales from the tail in (*A. laevis* only). A) Gross morphology of scale ~~sensillaorgans~~ on the nasal scale of *A.*
*laevis*. B) Gross morphology of the caudal scales of *A. laevis* illustrating sparse scale ~~sensillaorgans~~. C) Gross
morphology of scale ~~sensillaorgans~~ on the nasal scale of *H. stokesii*. Stereomicroscope images were taken from
museum specimens: A, B) WAMR174260 and C) FMNH202826. Scale bars represent 1 millimetre. Line
drawing based on image of *A. laevis* from (74)(74) and modified with permission.

**Figure 2.** Light micrographs of a ~~transverse section of~~ cephalic skin (supralabial scale) from *Aipysurus laevis*.
573 A) ~~Transverse section shows that Scale-scale sensillaorgans~~ (*) are skin elevations (bumps) created by dermal
papillae; ~~(dermal capsules)~~; other features of the dermis are clearly visible including nerve bundles, blood vessels
and collagen; note that the beta layer has artificially separated from alpha layer. B-C) Higher magnification of
transverse ~~eross~~ section of scale ~~sensillaorgans~~ (*) that show central cells within the dermal ~~capsulepapilla~~,
which displace the *stratum germinativum* of the epidermis; dermal ~~capsulespapillae~~ are vascularised by blood
vessels; note the red blood cells (rbc), lamellar corpuscles (lc) and melanophores (m) within the dermis. Slides
were stained with ~~hematoxylin~~ ~~hematoxylin~~-eosin and magnified at A) $\times 5.5$, B) $\times 20$ and C) $\times 30$.

**Figure 3.** Light micrographs of a ~~transverse section of~~ cephalic skin (nasal scale) from *Hydrophis stokesii*. A)
~~Transverse section shows that scale Scale-sensillaorgans~~ (*) are skin elevations (bumps) created by dermal
papillae; ~~(dermal capsules)~~; other features of the dermis are clearly visible including nerve bundles, blood vessels
and collagen fibres, and hinge region of the scale. B) Higher magnification of transverse ~~eross~~ section of scale
~~sensillumorgan~~ (*), the central cells within the dermal ~~capsulepapilla~~ displace the *stratum germinativum* of the
epidermis. C) ~~Transverse eross~~ section of edge of scale ~~sensillumorgan~~ shows a small bundle of collagen fibres

surrounded by central cells. Note the lamellar corpuscles (lc) within the dermis. Slides were stained with
Gomori's one-step and magnified at A) $\times 6.2$, B) $\times 22.8$ and C) $\times 23$.

**Figure 4.** Light micrographs of a transverse ~~eross~~-section of cephalic skin (nasal scale) ~~of from~~ *Hydrophis*
*stokesii* showing ~~that~~ dermal ~~eapsulepapillae~~ are not associated with external skin elevations (bumps). A-B)
Central cells of a dermal ~~eapsulepapillae~~ (*) displace surrounding *stratum germinativum* of the epidermis, but
do not result in skin elevations. Nerve bundle are closely associated with base of the dermal ~~eapsulepapilla~~.
Slide was stained with Gomori's one-step and magnified at A) $\times 20$ and B) $\times 40.7$

**Figure 5.** Immuno-reactivity of a neuron specific protein (PGP9.5) on cephalic skin (nasal scale) of *Hydrophis*
*stokesii*; reactive protein appears dark pink. A) Transverse cross-section of scale ~~sensillumorgan~~ (*) with
neuronal-positive stain within the dermal ~~eapsulepapillae~~, as well as within the epidermis and alpha layer above
the dermal ~~eapsulepapillae~~. Several neuronal-positive, discoid endings (arrows) are present within the *stratum*
*germinativum* and alpha layers of the epidermis. Lamellar corpuscles (lc) within the dermis are also immuno-
positive and can be distinguished from melanocytes (me) and dispersed melanophores (m), which have a dark
brown colouration. B) ~~Deeper-cross-Transverse~~ sections of a scale ~~sensillumorgan~~ showing neuronal-positive
discoid endings (arrows). C) A trail of neuronal-positive stain (arrow heads) leading to a forming scale
~~sensillumorgan~~ (*). Negative control was conducted by omitting primary antibody. Slides were counter stained
with Harris hematoxylin and magnified at A) $\times 30$, B) $\times 50$, C) $\times 50$.

**Figure 6.** Immuno-reactivity of a neuron specific protein (PGP9.5) of lamellar corpuscles (lc) in the cephalic
dermis (nasal scale) of *Hydrophis stokesii*. The location within the dermis, and co-localisation of immuno-
staining with lamellar structures suggests that they are Pacinian-like corpuscles. A) Immuno-reactivity of
PGP9.5, reactive protein appears dark pink, showing immuno-positive stain localised to lamellar corpuscles (lc)
in the dermis and discoid endings (arrows) in the epidermis. These structures can be distinguished from
melanocytes (me) and dispersed melanophores (m), which have a dark brown colouration. B) ~~Transverse-Deeper~~
~~eross~~-sections of the skin showing structure of lamellar corpuscles and an associated blood vessel (bv) and
nerve bundle (n). Slides were stained and magnified: A) Harris hematoxylin, $\times 30$, and B) ~~Gomori's One~~
~~StepGomori's one step trichrome~~, $\times 50$.

**Figure 7.** Light micrograph and transmission electron micrographs (TEM) of cross sections of cephalic scale
~~sensillaorgans~~ in sea snakes. A) Transverse cross-section of scale ~~sensillaorgans~~ (*) in *Aipysurus laevis* showing
the dermal ~~eapsulepapilla~~ within the epidermis. B) Higher magnification of dermal ~~eapsulepapilla~~ (*) in *A.*
*laevis*. First inset shows nuclei of central cells (c) and epidermal cells (keratinocytes; k), and collagen fibres
(coll), a structure typically found within the dermis, in the intercellular domain of the dermal ~~eapsulepapilla~~. A
putative myelinated axon (arrow heads) is present in the intercellular domain of the central cells; small
phospholipid (p) inclusions are also present. Inset two shows intercellular junctions (desmosomes; d) at the
membrane of central cells (c) and the keratinocytes (k). Note the fine ~~keratin-like tonofibrils~~ ~~keratin filaments~~
~~(tonofibrils; (tb))~~ associated with the desmosomes and large aggregations of keratin-~~like~~ filaments
~~(tonofilaments; (t))~~ in the intracellular domain of the keratinocytes. Light micrograph slide was stained with
hemotoxylin-eosin and magnified at A) $\times 34.1$; TEM: B) $\times 1900$, 1a) $\times 4800$, 1b) $\times 6800$; 3a) $\times 9300$ and 3b)
$\times 18,500$.

**Figure 8.** Light micrographs of transverse cross-sections of tail skin (posterior caudal scales) of *Aipysurus*
*laevis*. A) SA-scale sensillaorgans (*) in the tail are skin elevations created by a thickening of underlying
epidermis. B) Unknown dermal eapsulepapillae (*) consists of central cells that displaces surrounding *stratum*
*germinativum* of the epidermis, but does not result in skin elevations. Note that the dermis immediately
underlying dermal eapsulepapillae consists of loosely arranged collagen fibres devoid of melanophores (m).
Slides were stained with hemotoxylinhematoxylin-eosin and magnified at A) $\times 16$, B) $\times 17.2$

**References**

- 1. Cheng C, Wu P, Baker RE, Maini PK, Alibardi L, Chuong C-M. Reptile scale
paradigm: Evo-Devo, pattern formation and regeneration. *Int J Dev Biol.*
2010;53:813–26.
- 2. Lillywhite HB, Maderson PF. The structure and permeability of integument. *Am Zool.*
1988;28(3):945–62.
- 3. Maderson PFA, Rabinowitz T, Tandler B, Alibardi L. Ultrastructural contributions to
an understanding of the cellular mechanisms involved in lizard skin shedding with
comments on the function and evolution of a unique lepidosaurian phenomenon. *J*
*Morphol.* 1998;236(1):1–24.
- 4. Dehnhardt G, Mauck B. The physics and physiology of mechanoreception. In:
Nummela S, Thewissen JGM, editors. *Sensory evolution on the threshold: adaptations*
*in secondarily aquatic vertebrates.* Berkeley, CA: University of California Press; 2008.
- 5. Povel D, VanDerKooij J. Scale sensillae of the file snake (Serpentes: Acrochordidae)
and some other aquatic and burrowing snakes. *Netherlands J Zool.* 1997;47:443–56.
- 6. Underwood G. Characters useful in the classification of snakes. In: *A contribution to*
*the classification of snakes.* London: Trustees of The British Museum (Natural
History); 1967. p. 5–57.
- 7. Crowe-Riddell JM, Snelling EP, Watson AP, Suh AK, Partridge JC, Sanders KL. The
evolution of scale sensilla in the transition from land to sea in elapid snakes. *R Soc*
*Open Biol* [Internet]. 2016 Jun 8;6:160054. Available from:
<http://rsob.royalsocietypublishing.org/content/6/6/160054.abstract>
- 8. Keathley VL. *Tactile discrimination in three species of garter snake (Thamnophis).*
The University of Texas at Arlington. The University of Texas at Arlington; 2004.
- 9. Young BA, Morain M. Vertical burrowing in the Saharan Sand Vipers (*Cerastes*).
*Copeia.* 2003;2003(1):131–7.
- 10. Noble GK. The sense organs involved in the courtship of *Storeria*, *Thamnophis* and
other snakes. *Bull Am Museum Nat Hist.* 1937;73:673–725.
- 11. Nishida Y, Yoshie S, Fujita T. Oral sensory papillae, chemo- and mechano-receptors,
in the snake, *Elaphe quadrigata*. A light and electron microscopic study. Vol. 63,
*Archives of histology and cytology.* 2000. p. 55–70.
- 12. Aota S. An histological study on the integument of a blind snake, *Typhlops braminus*
(Daudin), with special reference to the sense organs and nerve ends. *J Sci Hiroshima*

- Univ. 1940;7:193–208.
- 13. Simões BF, Sampaio FL, Douglas RH, Kodandaramaiah U, Casewell NR, Harrison
RA, et al. Visual pigments, ocular filters and the evolution of snake vision. *Mol Biol*
*Evol* [Internet]. 2016;33(10):msw148. Available from:
<http://mbe.oxfordjournals.org/lookup/doi/10.1093/molbev/msw148>
- 14. Young BA. Snake bioacoustics: toward a richer understanding of the behavioral
ecology of snakes. *Q Rev Biol*. 2003;78(3):303–25.
- 15. Lillywhite HB. Perceiving the snake’s world. In: *How snakes work: structure,*
*function, and behavior of the world’s snakes.* Oxford University Press; 2014. p. 163–
79.
- 16. Gracheva EO, Ingolia NT, Kelly YM, Cordero-Morales JF, Hollopeter G, Chesler AT,
et al. Molecular basis of infrared detection by snakes. *Nature* [Internet].
2010;464(7291):1006–11. Available from:
<http://www.nature.com/doi/10.1038/nature08943>
- 17. Jackson MK, Doetsch GS. Functional properties of nerve fibers innervating cutaneous
corpuscles within cephalic skin of the Texas rat snake. *Exp Neurol*. 1977;56:63–77.
- 18. Proske U. Vibration-sensitive mechanoreceptors in snake skin. *Exp Neurol*.
1969;232:187–94.
- 19. Proske U. An electrophysiological analysis of cutaneous mechanoreceptors in a snake.
*Comp Biochem Physiol*. 1969;29:1039–46.
- 20. Proske U. Nerve endings in skin of the Australian black snake. *Anat Rec*.
1969;164(259–266):259–65.
- 21. von Düring M, Miller MR. Sensory nerve endings of the skin and deeper structures. In:
Gans C, editor. *Biology of the Reptilia.* New York: Academic Press; 1979. p. 407–41.
- 22. Jackson MK. Histology and distribution of cutaneous touch corpuscles in some
Leptotyphlopidae and Colubrid Snakes (Reptilia, Serpentes). *J Herpetol*. 1977;11(1):7–
15.
- 23. Jackson MK, Sharawy M. Scanning electron microscopy and distribution of
specialized mechanoreceptors in the Texas rat snake, *Elaphe obsoleta lindheimeri*. *J*
*Morphol*. 1980;163:59–67.
- 24. Roudaut Y, Lonigro A, Coste B, Hao J, Delmas P, Crest M. Touch sense: functional
organization and molecular determinants of mechanosensitive receptors. Vol. 6,
*Channels*. 2012. p. 234–45.
- 25. Zimmerman A, Bai L, Ginty DD. The gentle touch receptors of mammalian skin.

- Science (80-). 2014;346(6212):950–4.
- 26. Avolio C, Shine R, Pile A. Sexual dimorphism in scale rugosity in sea snakes
(Hydrophiidae). Biol J Linn Soc. 2006;89(2):343–54.
- 27. Underwood G. Characters useful in the classification of snakes. In: A contribution to
the classification of snakes. London: The British Museum (Natural History); 1967. p.
41–4.
- 28. Young BA, Wallach V. Description of a papillate tactile organ in the Typhlopidae.
South African J Zool. 1998;33(October):249–53.
- 29. Wallach V, Ineich I. Redescription of a rare Malagasy blind snake, *Typhlops*
*grandidieri* Mocquard, with Placement in a New Genus (Serpentes: Typhlopidae). J
Herpetol [Internet]. 1996;30(3):367–76. Available from:
<http://www.jstor.org/stable/1565174>
- 30. Sanders KL, Lee MSY, Leys R, Foster R, Keogh JS. Molecular phylogeny and
divergence dates for Australasian elapids and sea snakes (hydrophiinae): evidence
from seven genes for rapid evolutionary radiations. J Evol Biol. 2008;21(3):682–95.
- 31. Leitch DB, Catania KC. Structure, innervation and response properties of
integumentary sensory organs in crocodylians. J Exp Biol [Internet].
2012;215(23):4217–30. Available from:
<http://jeb.biologists.org/content/215/23/4217.abstract>
- 32. Jackson MK, Butler DG, Youson JH. Morphology and ultrastructure of possible
integumentary sense organs in the estuarine crocodile (*Crocodylus porosus*). J
Morphol. 1996;229:315–24.
- 33. Soares D. An ancient sensory organ in crocodylians. Nature. 2002;417:241–2.
- 34. Catania KC, Leitch DB, Gauthier D. Function of the appendages in tentacled snakes
(*Erpeton tentaculatus*). J Exp Biol. 2010;213:359–67.
- 35. Zimmerman K, Heatwole H. Cutaneous photoreception: a new sensory mechanism for
reptiles. Copeia. 1990;1990(3):860–2.
- 36. Riddell JMC, Hunt DM, Simões BF, Partridge JC, Delean S, Schwerdt JG, et al.
Phototactic tails: Evolution and molecular basis of a novel sensory trait in sea snakes.
Mol Ecol. 2019;00:1–16.
- 37. Avolio C, Shine R, Pile A. The adaptive significance of sexually dimorphic scale
rugosity in sea snakes. Am Nat [Internet]. 2006;167(5):728–38. Available from:
<http://www.jstor.org/stable/pdfplus/10.1086/503386.pdf>
- 38. Dean B, Bhushan B. Shark-skin surfaces for fluid-drag reduction in turbulent flow: a

- review. *Philos Trans R Soc London A Math Phys Eng Sci.* 2010;368(1929):4775–806.
- 39. Fish FE, Weber PW, Murray MM, Howle LE. The tubercles on humpback whales’
flippers: Application of bio-inspired technology. *Integr Comp Biol.* 2011;51(1):203–
13.
- 40. Gomori G. A rapid one-step trichrome stain. *Am J Clin Pathol.* 1950;20:661–4.
- 41. null null. R: A Language and Environment for Statistical Computing. 2017.
- 42. Matveyeva TN, Ananjeva NB. The distribution and number of the skin sense organs of
agamid, iguanid and gekkonid lizards. *J Zool [Internet].* 1995;235(2):253–68.
Available from: <http://dx.doi.org/10.1111/j.1469-7998.1995.tb05142.x>
- 43. Ananjeva ANB, Dilmuchamedov ME, Matveyeva TN. The skin sense organs of some
Iguanid lizards. *J Herpetol.* 2010;25(2):186–99.
- 44. Stovall RH. Cephalic scale pits observed on the lizard, *Uta stansburiana*: light and
scanning electron microscopy. *J Herpetol.* 1985;19(3):425–8.
- 45. Landmann L. The sense organs in the skin of the head of Squamata (Reptilia). *Isr J*
*Zool [Internet].* 1975 Jan 1;24(3–4):99–135. Available from:
<https://www.tandfonline.com/doi/abs/10.1080/00212210.1975.10688416>
- 46. Cross PC, Mercer KL. Cell and tissue ultrastructure: A functional perspective. W. H.
Freeman and Company; 1993.
- 47. von Düring M. The ultrastructure of lamellated mechanoreceptors in the skin of
reptiles. *Z Anat Entwickl-Gesch.* 1973;143:81–94.
- 48. Lumpkin EA, Marshall KL, Nelson AM. The cell biology of touch. *J Cell Biol.*
2010;191(2):237–48.
- 49. Sherratt E, Rasmussen AR, Sanders KL. Trophic specialization drives morphological
evolution in sea snakes. *R Soc Open Sci.* 2018;5(3):172141.
- 50. Heatwole H. Food and feeding. In: *Sea Snakes.* Hong Kong: UNSW Press; 1999. p.
46–50.
- 51. Westhoff G, Fry BG, Bleckmann H. Sea snakes (*Lapemis curtus*) are sensitive to low-
amplitude water motions. *Zoology.* 2005;108:195–200.
- 52. Budelmann BU, Bleckmann H. A lateral line analogue in cephalopods: water waves
generate microphonic potentials in the epidermal head lines of *Sepia* and *Lolliguncula*.
*J Comp Physiol A.* 1988;164:1–5.
- 53. Coombs S, Janssen J, Webb JF. Diversity of lateral line systems: evolutionary and
functional considerations. In: Atema J, Fay RR, Popper AN, Tavolga WN, editors.
*Sensory biology of aquatic animals.* New York: Springer-Verlag; 1987. p. 553–93.

- 54. Gläser N, Wieskotten S, Otter C, Dehnhardt G, Hanke W. Hydrodynamic trail
following in a California sea lion (*Zalophus californianus*). *J Comp Physiol A*
[Internet]. 2011;197(2):141–51. Available from: [http://dx.doi.org/10.1007/s00359-](http://dx.doi.org/10.1007/s00359-010-0594-5)
010-0594-5
- 55. Hanke W, Wieskotten S, Marshall C, Dehnhardt G. Hydrodynamic perception in true
seals (Phocidae) and eared seals (Otariidae). *J Comp Physiol*. 2012/11/28.
2013;199(6):421–40.
- 56. Dehnhardt G, Mauck B, Hanke W, Bleckmann H. Hydrodynamic trail-following in
harbor seals (*Phoca vitulina*). *Science* (80-) [Internet]. 2001;293(5527):102–4.
Available from: <http://www.sciencemag.org/content/293/5527/102.abstract>
- 57. Winokur RM. The integumentary tentacles of the snake *Erpeton tentaculatum*:
structure, function, evolution. *Herpetologica*. 1977;33(2):247–53.
- 58. Catania KC. Tactile sensing in specialized predators - from behavior to the brain. *Curr*
*Opin Neurobiol* [Internet]. 2012;22(2):251–8. Available from:
<http://dx.doi.org/10.1016/j.conb.2011.11.014>
- 59. Voris HK. Fish eggs as the apparent sole food item for a genus of sea snake,
*Emydocephalus* (Krefft). *Ecology*. 1966;47(1):152–4.
- 60. Pisani GR. Comments on the courtship and mating mechanics of *Thamnophis*
(Reptilia, Serpentes, Colubridae). *J Herpetol*. 2012;10(2):139–42.
- 61. Northcutt GR. The phylogenetic distribution and innervation of craniate
mechanoreceptive lateral lines. In: Coombs S, Görner P, Münz H, editors. *The*
*mechanosensory lateral line*. New York, NY: Springer; 1989. p. 17–78.
- 62. Di-Poi N, Milinkovitch MC. Crocodylians evolved scattered multi-sensory micro-
organs. *Evodevo* [Internet]. 2013;4(1):19. Available from:
[http://www.pubmedcentral.nih.gov/articlerender.fcgi?artid=3711810&tool=pmcentrez](http://www.pubmedcentral.nih.gov/articlerender.fcgi?artid=3711810&tool=pmcentrez&rendertype=abstract)
[&rendertype=abstract](http://www.pubmedcentral.nih.gov/articlerender.fcgi?artid=3711810&tool=pmcentrez&rendertype=abstract)
- 63. Brooks D, Jackson K. Do crocodiles co-opt their sense of “touch” to “taste”? A
possible new type of vertebrate sensory organ. *Amphibia-Reptilia* [Internet].
2007;28(2):277–85. Available from:
<http://booksandjournals.brillonline.com/content/10.1163/156853807780202486>
- 64. Baker GE, de Grip WJ, Turton M, Wagner H-J, Foster RG, Douglas RH. Light
sensitivity in a vertebrate mechanoreceptor? *J Exp Biol*. 2015;218(18):2826–9.
- 65. Pei X, Wilkens LA, Moss F. Light enhances hydrodynamic signaling in the
multimodal caudal photoreceptor interneurons of the crayfish. *J Neurophysiol*

- [Internet]. 1996;76(5):3002–11. Available from:
<http://jn.physiology.org/content/76/5/3002.abstract>
- 66. Bennett MVL. Electroreceptors. In: Hoar WS, Randall DJ, editors. *Fish Physiology*.
New York, NY: Academic Press; 1971. p. 493–574.
- 67. Szabo T. Sense organs of the lateral line system of some electric fish of the
Gymnotidae, Gymnarchidae, and Mormyridae. *J Morphol*. 1965;117:229–50.
- 68. Brisichoux F, Tingley R, Shine R, Lillywhite HB. Salinity influences the distribution of
marine snakes: implications for evolutionary transitions to marine life. *Ecography*
(Cop). 2012;35:994–1003.
- 69. Lillywhite HB, Sheehy CM, Sandfoss MR, Crowe-Riddell JM, Grech A. Drinking by
sea snakes from oceanic freshwater lenses at first rainfall ending seasonal drought.
*PLoS One*. 2019;14(2):1–11.
- 70. Lillywhite HB, Babonis LS, Sheehy CM, Tu M-C. Sea snakes (*Laticauda* spp.) require
fresh drinking water: implication for the distribution and persistence of populations.
*Physiol Biochem Zool* [Internet]. 2008;81(6):785–96. Available from:
<http://www.journals.uchicago.edu/doi/abs/10.1086/588306>
- 71. Lillywhite HB, Heatwole H, Sheehy CMI. Dehydration and drinking behavior in true
sea snakes (Elapidae: Hydrophiinae: Hydrophiini). *J Zool*. 2012;296(1).
- 72. Burns B. Oral sensory papillae in sea snakes. *Copeia*. 1969;1969(3):617–9.
- 73. Jackson MK, Doetsch GS. Response properties of mechanosensitive nerve fibers
innervating cephalic skin of the Texas rat snake. *Exp Neurol*. 1977;56:78–90.
- 74. Sanders KL, Somaweera R. *Guide to the Sea Snakes of the Kimberley Coast of*
*Western Australia*.